# CENSORING WITH PLAUSIBLE DENIABILITY: ASYMMETRIC LOCAL PRIVACY FOR MULTI-CATEGORY CDF ESTIMATION

## ABSTRACT

We introduce a new mechanism within the Utility-Optimized Local Differential Privacy (ULDP) framework that enables censoring with plausible deniability when collecting and analyzing sensitive data. Our approach addresses scenarios where certain values, such as large numerical responses, are more privacy-sensitive than others, while accompanying categorical information may not be private on its own but could still be identifying. The mechanism selectively withholds identifying details when a response might indicate sensitive content, offering asymmetric privacy protection. Unlike previous methods, it avoids the need to predefine which values are sensitive, making it more adaptable and practical. Although the mechanism is designed for ULDP, it can also be applied under symmetric LDP settings, where it still benefits from censoring and reduced privacy cost. We provide theoretical guarantees, including uniform consistency and pointwise weak convergence results. Extensive numerical experiments demonstrate the validity of developed methodologies.

## 1 INTRODUCTION

Although crowd-sourced data aggregation has led to impressive large-scale telemetry-driven services, such as Google Maps and Apple's predictive keyboard, collecting statistics from personal data while preserving individual privacy remains a fundamental challenge in the age of big data. Differential Privacy (DP) (Dwork et al., 2006a), has become the prevailing standard for privacy-preserving analysis recognized by its notable deployments in the U.S. Census Bureau's 2020 Census (Hotz & Salvo, 2022; Abowd & Hawes, 2023). Although DP (or central-DP for contrast), controls the leakage of privacy at publication, it is vulnerable towards curator side breaches (see Ayyagari (2012); Quach et al. (2022); Lee (2022); Khan et al. (2022); Hantke et al. (2024) for such incidents and studies). Such events call for protection that is closer to its source, the protocol of data collection. Local Differential Privacy (LDP) (Duchi et al., 2013) has emerged as a powerful alternative. By removing the need for a trusted curator, LDP allows users to locally randomize their responses, ensuring that even the data collector cannot infer sensitive information with high confidence. LDP protocols have been widely adopted for collecting privatized data, including major companies like Tiktok (TikTok Engineering, 2023), Google (Erlingsson et al., 2014) and Microsoft (Ding et al., 2017).

While LDP offers robust privacy guarantees, it inherently imposes a non-negligible utility loss, even under optimal conditions (Steinberger, 2024). This trade-off is already evident in the basic task of frequency estimation (Wang et al., 2017). The situation is exacerbated when multiple attributes are collected (Liu et al., 2023; Arcolezi et al., 2023): either by combining the attributes into a high-dimensional domain, which increases the complexity and potential error, or by splitting the privacy budget among attributes, leading to reduced accuracy for each. Both approaches have been shown to significantly degrade estimation accuracy in such settings. Considering this, it is natural to ask whether we must pay for all aspects of privacy equally—or whether we can instead choose what to protect. We begin by observing that sensitivity is often asymmetric in numerical attributes. For instance, high income may be considered sensitive due to tax or benefit implications, while low income is less concerning. Similarly, high debt or frequent insurance claims may reveal undesirable traits, whereas low values are relatively innocuous. In other domains, the opposite is true—such as GPA, where low values are more embarrassing or private. At the same time, many real-world surveys

also include categorical demographic attributes such as nationality, gender, or postal code to support fairness, subgroup analysis, or other stratified inference tasks. Naively applying LDP to such data requires protecting every attribute equally, effectively treating nonsensitive fields as if they are just as sensitive. Related relaxations under LDP show that substantial utility gains are possible, as in the semi-feature LDP framework for nonparametric regression with public features Ma et al. (2025), but attributes that are not sensitive on their own can act as quasi-identifiers (Borrero-Foncubierta et al., 2025; Wong et al., 2019). This is especially true for continuous nonsensitive variables (e.g., precise income or debt), which may be nearly unique and thus indirectly revealing. Adding noise uniformly wastes privacy budget or discourages truthful reporting when users are unwilling to disclose sensitive data with identifiers. One solution is defining a fixed sensitive region (Murakami & Kawamoto, 2019): define a region of sensitivity and allow disclosure of values outside. But such regions can be arbitrary, vary across individuals, and shift over time. In contrast, the direction of sensitivity tends to be more stable—for example, it is much more likely that higher debt is sensitive than low.

## 1.1 RELATED WORKS

Without the constraint of DP, empirical cumulative distribution functions can already be close to the underlying truth; such studies may date back to Komlós et al. (1975). On the central model, where a trusted curator has access to raw data, various mechanisms have been proposed for accurate distribution estimation. Barber & Duchi (2014) demonstrated that histogram estimators are optimal for Lipschitz distributions under the $L_2$ risk in the presence of differential privacy constraints. Later, Lalanne et al. (2023) extended this work by analyzing the cost of central privacy in estimating the density of densities in the Lipschitz and Sobolev spaces. Beyond distributional functionals, versatile central mechanisms based on data-level perturbations—such as the zero-inflated multivariate Laplace mechanism for general M-estimation proposed by Lu et al. (2025), illustrating how carefully designed noise distributions can support a broad class of privacy guarantees.

In the context of LDP, the estimation of distributions over continuous domains presents unique challenges. For discrete domains, frequency oracle mechanisms such as RAPPOR (Erlingsson et al., 2014) and Hadamard Response (Acharya et al., 2019) have been developed. These methods can be extended to continuous data through discretizations, but this approach may compromise the inherent structure of the continuous domain. To better preserve the characteristics of continuous data, several LDP perturbation techniques have been proposed. These include the direct application of the Laplace mechanism (Dwork et al., 2006b), the piecewise mechanism (Wang et al., 2019), its refinement for improved utility (Li et al., 2020), and more recently, a binary response-based approach (Liu et al., 2024). These methods aim to balance the trade-off between privacy and accuracy, particularly under the constraints of limited information channels inherent to LDP.

The concept of Utility-Optimized Local Differential Privacy (ULDP) was introduced in Murakami & Kawamoto (2019), initially for frequency estimation via modified randomized response and RAPPOR-style mechanisms. This framework aims to enhance utility by allowing users to specify sensitive regions, thereby relaxing the privacy constraints on non-sensitive data. Subsequent work (Zhang et al., 2024) has extended ULDP to the $(\epsilon, \delta)$ setting, with a refined control on privacy leakage, and Zhang et al. (2024) proposed mean estimation techniques for numerical data in the same framework, allowing robust private aggregation of continuous values. To the best of our knowledge, however, there has been no prior work addressing the estimation of distribution of a continuous variable—whether standalone or paired with a categorical demographic attribute—under utility-optimized LDP.

## 1.2 OUTLINE

We begin by reviewing the relevant definitions and background on CDF estimation and differential privacy frameworks. This is followed by a description of our data collection procedure, which employs a deterministic preprocessing step that maps the original secret information to a binary response—similar in spirit to Liu et al. (2024), but without introducing random perturbation at this stage. The resulting binary response is then processed through a randomized ULDP mechanism. This two-step design avoids the need to predefine a sensitive region and instead only specifies the direction of sensitivity (e.g., larger values are considered sensitive).

Next, we construct an estimator based on the privatized binary data. A key observation is that the privacy mechanism and the statistical estimation procedure can be cleanly separated by adopting

an alternative interpretation of the randomized response: it can be viewed as a truthful response from a transformed variable, akin to techniques in Liu et al. (2024) and conceptually similar to data encountered in competing risks settings in medical statistics. Building on this insight, we develop a maximum likelihood estimator (MLE) by discretizing the data and solving a bound-constrained optimization problem, resulting in an estimator with a data-driven support.

We establish the $L_2$- and $L_\infty$- uniform consistency of the ULDP CDF estimator under a general privacy mechanism, with convergence rates of $\mathcal{O}_p(n^{-1/3})$, $\mathcal{O}_p(n^{-1/3} \log n)$ respectively. Furthermore, we derive pointwise weak convergence results at interior points. These findings are consistent with the results obtained for the case $K = 1$ under the LDP mechanism studied in Liu et al. (2024), which rely on the Chernoff distribution properties described in Groeneboom (1989). Building on the ULDP CDF estimator, we also demonstrate how to construct consistent estimators for the predictive probabilities of categorical outcomes, conditioned on a given range of sensitive features. To the best of our knowledge, this is the first work to establish these asymptotic properties for ULDP CDF estimation and its application to multi-category prediction.

Finally, we discuss the implementation details of the algorithm and validate the effectiveness of our proposed protocol through numerical experiments, demonstrating its practical utility and accuracy in estimating the CDF under the ULDP framework.

The remainder of the paper is organized as follows. Section 2 introduces the background on differential privacy frameworks. Section 3 describes the problem setting and methodology. Section 4 establishes the asymptotic properties of the proposed estimator, and Section 5 investigates its finite sample performance. Additional discussions, simulation results and all technical proofs are provided in the Appendix.

## 2 Preliminaries

### 2.1 Differential Privacy: Central and Local Models

Differential Privacy (DP) provides a rigorous framework for protecting individual information in data analysis. At its core, DP ensures that the output of a computation remains statistically indistinguishable whether or not any one individual's data is included. This protects against inference attacks, even by adversaries with substantial auxiliary knowledge.

**Definition 1 (Dwork et al., 2006a)** *A randomized mechanism $\mathcal{A}$ is $(\epsilon, \delta)$-differentially private if, for all datasets $S, S'$ differing on a single individual's data and all measurable subsets $E$ of outputs,*

$$\mathbb{P}[\mathcal{A}(S) \in E] \leqslant e^\epsilon \mathbb{P}[\mathcal{A}(S') \in E] + \delta.$$

In the central DP (CDP) model, this guarantee is enforced by a trusted data curator who aggregates the dataset and injects noise into the final output. While CDP typically yields high utility, it assumes users trust the curator with their raw data.

In contrast, the local models remove the need for trust: each user independently applies a randomization mechanism to their data before sharing it. The formal definition is as follows:

**Definition 2 (Joseph et al., 2019)** *A randomized mechanism $R : \mathcal{X} \to \mathcal{Y}$ satisfies $(\epsilon, \delta)$-LDP if, for all inputs $x, x' \in \mathcal{X}$ and measurable subsets $S \subseteq \mathcal{Y}$,*

$$\mathbb{P}[R(x) \in S] \leqslant e^\epsilon \mathbb{P}[R(x') \in S] + \delta.$$

In LDP, each user has full control over their privacy, and no trusted aggregator is required. However, the noise introduced at the individual level often imposes a high utility cost—particularly when estimating fine-grained statistics or when multiple attributes must be protected.

### 2.2 Utility-Optimized Local Differential Privacy

To mitigate the utility degradation under LDP, utility-optimized local differential privacy (ULDP) was proposed, initially for categorical distribution estimation (Murakami & Kawamoto, 2019). ULDP

provides strong privacy guarantees only over a predefined sensitive region of the input domain while allowing exact outputs for the non-sensitive region when it does not risk user privacy. The formal definition is as follows:

**Definition 3 (Murakami & Kawamoto, 2019)** *A randomized mechanism* $\mathbf{Q} : \mathcal{X} \to \mathcal{Y}$ *satisfies* $(\mathcal{X}_S, \mathcal{Y}_P, \epsilon)$*-ULDP if:*

    *1. For any* $y \in \mathcal{Y}_I := \mathcal{Y} \backslash \mathcal{Y}_P$, *there exists* $x \in \mathcal{X}_N := \mathcal{X} \backslash \mathcal{X}_S$ *such that*

$$\mathbf{Q}\left(y \mid x\right) > 0 \quad and \quad \mathbf{Q}\left(y \mid x'\right) = 0 \text{ for all } x' \neq x.$$

    *2. For any* $x, x' \in \mathcal{X}$ *and any* $y \in \mathcal{Y}_P$, $\mathbf{Q}\left(y \mid x\right) \leqslant e^{\epsilon} \mathbf{Q}\left(y \mid x'\right)$.

This definition is slightly generalized in Zhang et al. (2024) to allow a continuous output space $\mathcal{Y}$ and relaxed probabilistic guarantees. However, we adopt the original discrete formulation, as these generalizations are not relevant to our setting.

In this definition, $\mathcal{X}_S$ denotes the sensitive subset of the input domain, and $\mathcal{Y}_P$ represents the subset of outputs over which DP-style indistinguishability is enforced. For convenience, we refer to $\mathcal{X}_S$, $\mathcal{X}_N = \mathcal{X} \backslash \mathcal{X}_S$, $\mathcal{Y}_P$, and $\mathcal{Y}_I = \mathcal{Y} \backslash \mathcal{Y}_P$ as sensitive inputs, safe inputs, sensitive outputs, and safe outputs, respectively.

Notably, sensitive inputs never produce safe outputs. This design choice not only simplifies analysis and improves utility but also adds a safety guarantee because mapping to a safe but rare output may reveal it was perturbed from a sensitive input. Therefore, sensitive inputs always map to sensitive outputs, and perturbation occurs entirely within the sensitive output space (except in the degenerate case where there is only one sensitive output, in which case no perturbation is needed). Meanwhile, safe inputs may be mapped either to sensitive or safe outputs to provide plausible deniability for sensitive inputs.

## 3 METHODOLOGY

### 3.1 PROBLEM FORMULATION

We consider a population of $n$ users, each holding a data pair $(X, Y)$ drawn i.i.d. from an unknown joint distribution over $[0, 1] \times \{1, \ldots, K\}$. Here, $X$ is a numerical variable that may be sensitive, , $Y$ is a categorical variable that typically represents demographic information and $K$ is the number of categories. Without loss of generality, we assume $X \in [0, 1]$, with larger values of $X$ corresponding to increasingly sensitive information. In the mean time, our privacy goal follows the ULDP framework: any output that reveals or suggests that a user holds a larger value of $X$ must be protected by standard $\epsilon$-indistinguishability. That is, for any two inputs differing in $X$ their corresponding output distributions must remain within a multiplicative factor of $e^{\epsilon}$ for any sensitive output.

The utility goal is to estimate the joint distribution function $F_{0k}(t) = \mathbb{P}(X \leqslant t, Y = k)$ for each category $k = 1, \ldots, K$, which describes the cumulative distribution of $X$ conditioned on the categorical label. Estimation quality may be measured under various norms; in this work, we focus on the $L_{\infty}$ norm as a canonical metric for evaluating the maximum estimation error across the domain.

### 3.2 ULDP DATA COLLECTION

Unlike the CDP setting where raw data is collected and perturbed by a trusted aggregator, the design of the data collection procedure is crucial in the local setting. This is particularly challenging for continuous variables, where existing LDP or ULDP mechanisms such as additive noise (e.g., Laplace mechanism) or square wave encoding introduce large variance and often produce values outside the support, making recovery difficult (Fan, 1992).

In addition, direct application of ULDP requires the predefined specification of a sensitive region in $\mathcal{X}$, as done in Zhang et al. (2024). This requirement is at odds with the practical observation that sensitivity boundaries are difficult to determine and may vary over time. Predefining such regions rigidly may lead to inconsistent protection.

Motivated by recent works (Liu et al., 2024; Nikita & Steinberger, 2025), we adopt a binary encoding that bypasses the need for specifying a sensitive subset of $\mathcal{X}$ and improves estimation accuracy. In particular, each user is issued a threshold $t_i$, sampled from a random variable $T$ with a predetermined distribution $G$ over $[0, 1]$ (e.g., the uniform distribution). The user then compares their private value $x_i$ with the threshold $t_i$ and computes the binary indicator $\mathbf{1}_{x_i > t_i}$.

This comparison serves two purposes. First, the bit $\mathbf{1}_{x_i > t_i}$ is a function of $x_i$ and thus not more sensitive than $x_i$ itself—it can be computed from $x_i$ but not vice versa. Second, it introduces an asymmetry in sensitivity: reporting $\mathbf{1}_{x_i > t_i} = 1$ suggests that $x_i$ may be large and therefore sensitive, while $\mathbf{1}_{x_i > t_i} = 0$ does not indicate sensitive information and can be treated as non-sensitive while, eliminating the need to predefine a fixed sensitive zone in $\mathcal{X}$.

This structure also aligns well with the objectives of ULDP. Naively, after preprocessing with threshold comparison and combining with the categorical label $Y$, each user produces one of $2K$ possible outcomes—pairs $(\mathbf{1}_{x_i > t_i}, y_i)$, which we denote as $\mathcal{X} = \{0, 1\} \times \{1, \ldots, K\}$. Among these, half—those $\mathcal{X}_S = \{(1, y) | y \in \{1, \ldots, K\}\}$—are considered sensitive.

A standard LDP random response mechanism $\mathcal{A} : \mathcal{X} \to \mathcal{X}$ would be costly in terms of utility. The probability of returning the true value is

$$P(\mathcal{A}(x) = x) = \frac{e^\epsilon}{e^\epsilon + 2K - 1}.$$

For example, when $K = 4$ and $\epsilon = 1$, this results in a truthful response probability of less than $28\%$, with the remaining probability spread uniformly across the other $2K - 1 = 7$ outputs.

A direct application of the utility-optimized randomized response mechanism under ULDP (Murakami & Kawamoto (2019), Definition 3) yields truthful reporting probabilities of approximately $47\%$ for sensitive outputs and around $30\%$ for safe outputs under the same parameters (see F for details).[1] While these rates represent an improvement over standard LDP, they remain suboptimal.

To serve the dual purposes of utility and privacy, we propose suppressing the report of $Y$ for sensitive outputs entirely as follows:

**Definition 4 (Asymmetrically Censored Randomized Response (ACRR):)** *Given a privacy budget $\epsilon > 0$, define the output domain using one-hot encoding vectors as $\mathcal{E} = \{e_1, e_2, \ldots, e_K, e_{K+1}\} \subset \{0, 1\}^{K+1}$, where each $e_i$ is the one-hot vector with a 1 at the $i$-th position and 0 elsewhere.*

*The ACRR mechanism $\mathcal{M} : \mathcal{X} \to \mathcal{E}$ is defined as follows:*

$$\mathcal{M}((0, k)) = \begin{cases} e_k & \text{with probability } 1 - e^{-\epsilon}, \\ e_{K+1} & \text{with probability } e^{-\epsilon}, \end{cases} \quad \text{for } k = 1, \ldots, K,$$

*and $\mathcal{M}((1, k)) = e_{K+1}, \quad \text{for } k = 1, \ldots, K.$*

As a special case of the utility-optimized randomized response mechanism (Murakami & Kawamoto, 2019), this approach provides $(\mathcal{X}_N, \{e_{K+1}\}, \epsilon)$-ULDP. Numerically, for $\epsilon = 1$ and $K = 4$, the probability of truthful reporting is over $63\%$ for safe inputs and exactly $100\%$ for sensitive inputs. Notably, these probabilities remain independent of $K$ due to censoring, which further improves utility by eliminating the need for perturbation in the degenerate case of a singleton sensitive output set. To better illustrate the ACRR mechanism, we present a schematic diagram in Figure 1, and the full mechanism is provided in Appendix B.

**Remark 1** *An additional advantage of this approach is conditional censoring: if the output indicates that the numerical response is potentially sensitive ($x > t$), the mechanism suppresses disclosure of the accompanying categorical identifier $Y$. This conditional suppression reduces re-identification risks, enhancing privacy even further beyond the basic ULDP guarantees. See Appendix A for a real world example of a such survey.*

**Remark 2** *The ULDP guarantee applies to the derived space of $\mathcal{X}$ and $\mathcal{E}$. Due to the first condition in the original ULDP definition (Definition 1), a mapping from a continuous domain to a discrete*

---

[1]Safe inputs are less likely to be truthfully reported because they must be perturbed to provide plausible deniability for sensitive inputs.

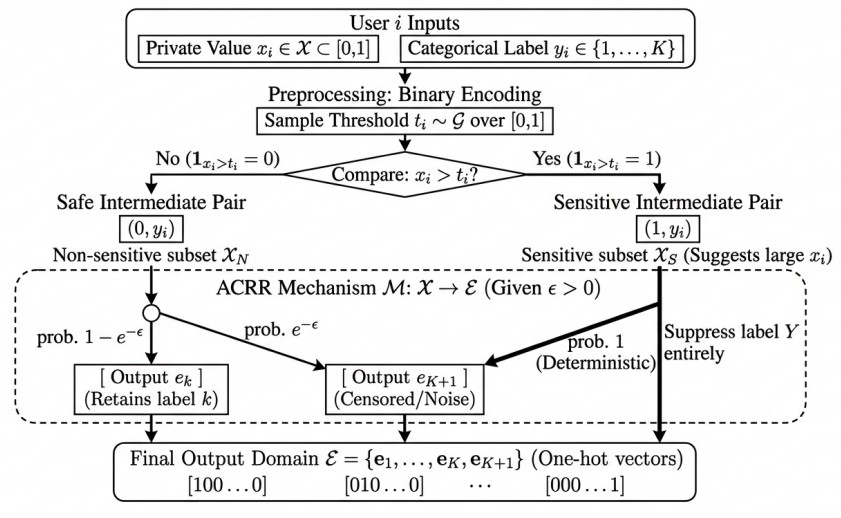

Figure 1: Schematic diagram for ACRR mechanism

*domain is not compatible with this requirement. Instead, we protect a coarsened representation that is sufficient for estimation and inherently less informative as an image of the original data. As with any mechanism under a local DP framework, the guarantee is fully disclosed to users to support informed decision-making.*

### 3.3 RECOVERING THE DISTRIBUTION FROM THE ULDP DATA VIEW

The ACRR provides an ULDP view of the data in the formation of one hot vectors of length $K + 1$. Next, we consider how to recover the original distribution from it.

To intuitively understand how recovery remains possible even under censoring mechanisms, let us temporarily remove randomness by setting $\epsilon = +\infty$. In this case, all indicator functions are true and all labels about $Y$ are true or missing.

Observe that for any threshold $u < 1$, the joint cumulative distribution function $F_{0k}(u) = \mathbb{P}(X \leqslant u, Y = k)$ can be estimated without requiring any information from the part of $X > u$.

Precisely, suppose that before applying the LDP mechanism, the data collected from user $i$ is summarized by

$$\Delta_i = (\Delta_{i,k}, 1 \leqslant k \leqslant K + 1) = (\mathbf{1}_{x_i \leqslant t_i, y_i = 1}, \ldots, \mathbf{1}_{x_i \leqslant t_i, y_i = K}, \mathbf{1}_{x_i > t_i}),$$

where $x_i$ is the continuous variable and $y_i \in \{1, \ldots, K\}$ is the categorical label.

Based on the observed data $\{\Delta_i\}_{i=1}^n$, we consider the following log-likelihood function for the distribution $\mathbf{F} = (F_{0k}, F_{1k}, \ldots, F_{0K})$ and $F_+ = \sum_{k=1}^K F_{0k}$.:

$$\ell(\mathbf{F}) = \sum_{i=1}^n \left( \sum_{k=1}^K \Delta_{i,k} \log F_{0k}(t_i) + \Delta_{i,K+1} \log(1 - F_+(t_i)) \right). \tag{1}$$

The estimation of each cumulative distribution function (CDF) $F_{0k}$ is obtained simultaneously by maximizing the log-likelihood function equation 1, subject to monotonicity constraints. Notice that each distribution $F_{0k}$ can be estimated individually using only the data $\{\Delta_{i,k}\}_{i=1}^n$, by maximizing the log-likelihood function

$$\ell(F_{0k}) = \sum_{i=1}^n \left( \Delta_{i,k} \log F_{0k}(t_i) + (1 - \Delta_{i,k}) \log(1 - F_{0k}(t_i)) \right). \tag{2}$$

However, this approach neglects the dependence structures among the distributions $F_{0k}$. Additionally, it is less efficient due to ignoring information contained in $\Delta_{i,K+1}$ (Maathuis & Hudgens, 2011).

This form of shape-constrained minimization surprisingly coincides with survival-censored data. In Hudgens et al. (2001), an Expectation-Maximization (EM) algorithm was proposed that transforms the MLE for truncated competing-risks data into an EM problem on the unknown interval-and-type allocations. Later an Iterative Convex Minorant (ICM) algorithm (Groeneboom & Jongbloed, 2014) was derived with more computational efficiency, which we will later adopt for experiments.

**Remark 3** *The idea of the ICM algorithm is to approximate the loss function using a weighted sum of squares and then perform iterative optimization by computing the left derivative of the convex minorant over a collection of points. Detailed descriptions of the algorithm can be found in Section 7.3 of Groeneboom & Jongbloed (2014) and we give a brief introduction in Appendix D Notably, the minimization of (1) leads to a step function on $t_i$ since the value elsewhere is irrelevant to the likelihood.*

Surprisingly, setting $\epsilon < \infty$ does not significantly complicate the recovery process.

After applying the LDP mechanism $\mathcal{M}$, we observe a perturbed indicator, which can be equivalently viewed as a sample from a new random variable $(X^\star, Y^\star)$ drawn from a distribution distorted by the mechanism. Accordingly, we define the transformed indicator

$$\Delta_i^\star = (\mathbf{1}_{x_i^\star \leqslant t_i,\, y_i^\star = 1}, \ldots, \mathbf{1}_{x_i^\star \leqslant t_i,\, y_i^\star = K},\, \mathbf{1}_{x_i^\star > t_i}),$$

as if the perturbed data were generated truthfully from $(X^\star, Y^\star)$. Let

$$F_{0k}^\star(t) = \mathbb{P}(X^\star \leqslant t,\, Y^\star = k), \quad \text{for } k = 1, \ldots, K,$$

and define the CDF vector under this distorted distribution as

$$\mathbf{F}^\star(t) = (F_{01}^\star(t), \ldots, F_{0K}^\star(t)), \quad F_+^\star(t) = \sum_{k=1}^{K} \mathbb{P}(X^\star \leqslant t,\, Y^\star = k)$$

In this formulation, the observed empirical estimates $\widehat{\mathbf{F}}^\star(t)$ can be computed from the data $\mathcal{E}(\Delta_i)$, and the original target CDF vector $\mathbf{F}(t)$ can be recovered by inverting the distortion introduced by the mechanism, which is where we pay the price of random perturbation, as the variance will be inflated in this procedure. We will quantify that in the next section. The complete procedure is summarized in the following algorithm.

---

**Algorithm 1** Estimation (server-side recovery of $F_{0k}$ from ACRR data)

---

**Inputs:** Collected reports $\{(t_i, a_i)\}_{i=1}^n$ with $a_i \in \{e_1, \ldots, e_K, e_{K+1}\}$. Parameters $K$(categories) and $\epsilon > 0$ (privacy budget).
**Outputs:** Step-function estimators $\{\widehat{F}_{0k}(t)\}_{k=1}^K$; $\widehat{F}_+(t) = \sum_{k=1}^K \widehat{F}_{0k}(t)$.
**Procedure:**

1. **Fit the distorted sub-CDFs.** Treat ACRR outputs as current-status competing risks; estimate nondecreasing step functions $\{\widehat{F}_{0k}^\star(t)\}_{k=1}^K$ on the thresholds using a monotone method (e.g., likelihood/ICM), and set $\widehat{F}_+^\star(t) = \sum_{k=1}^K \widehat{F}_{0k}^\star(t)$.

2. **Reverse DP.** Using $F_{0k}^\star(t) = (1 - e^{-\epsilon}) F_{0k}(t)$, set

$$\widehat{F}_{0k}(t) \leftarrow \frac{\widehat{F}_{0k}^\star(t)}{1 - e^{-\epsilon}}, \qquad \widehat{F}_+(t) \leftarrow \sum_{k=1}^{K} \widehat{F}_{0k}(t).$$

3. **Postprocess (total-CDF capping).** Cap the total CDF at $1$ and freeze all sub-CDFs thereafter: $t^\dagger \leftarrow \inf\{t : \widehat{F}_+(t) \geqslant 1\}$ (set $t^\dagger = +\infty$ if empty), and for all $k$,

$$\widehat{F}_{0k}(t) \leftarrow \widehat{F}_{0k}(t)\mathbf{1}_{t \leqslant t^\dagger} + \widehat{F}_{0k}(t^\dagger)\mathbf{1}_{t > t^\dagger} \qquad , \widehat{F}_+(t) \leftarrow \min\{\widehat{F}_+(t), 1\}.$$

---

## 4 ASYMPTOTIC PROPERTIES

Beyond the ACRR mechanism, we prove the theoretical results on a broader family of censored perturbation mechanisms, including the mechanism in Liu et al. (2024), an LDP variant of ACRR (see Appendix F), which also benefits from the censoring. First, we define the censor map.

**Definition 5 (Censor map)** *The censor map $\mathcal{C} : \mathcal{X} \to \mathcal{E}$, where $(0, k)$ is mapped to $e_k$ for $k = 1, \dots, K$, and all other values are mapped to $e_{K+1}$.*

Then, any randomized mapping from $\mathcal{E}$ to $\mathcal{E}$ can be represented by a $(K + 1) \times (K + 1)$ transition matrix $\mathcal{L}$, where $\mathcal{L}_{i,j}$ denotes the probability that $e_i$ is mapped to $e_j$. Let $\mathcal{W}_{\mathcal{L}}$ denote the randomization mechanism induced by $\mathcal{L}$. Then, the ACRR mechanism can be expressed as the composition of $\mathcal{W}_{\mathcal{L}}$ and the censoring map $\mathcal{C}$, where $\mathcal{L}$ is defined as follows:

$$\begin{bmatrix} (1 - e^{-\epsilon})I_K & e^{-\epsilon} \cdot \mathbf{1}_K \\ \mathbf{0}_{1 \times K} & 1 \end{bmatrix}.$$

**Remark 4** *It is worth noting that the relationship between the true distribution $\mathbf{F}(t)$ and the observed distribution $\mathbf{F}^{\star}(t)$ is $\mathbf{F}^{\star}(t) = \mathcal{L} \cdot \mathbf{F}(t)$, and recovery amounts to computing $\mathbf{F}(t) = \mathcal{L}^{-1}\mathbf{F}^{\star}(t)$, provided $\mathcal{L}$ is invertible. This framework accommodates a broad class of mechanisms beyond simple randomized response and allows for principled estimation under LDP with finite $\epsilon$.*

While it may appear natural to extend existing uniform consistency results for current-status data, this is nontrivial in our setting. The ULDP mechanism introduces additional non-differentiable points-privacy-induced artifacts that violate the smoothness assumptions underlying classical analyses. To handle this, we first establish key local properties of the estimator on neighborhoods that avoid these irregularities. We then use compactness, via the Heine–Borel theorem, to lift these local bounds to uniform control over the full interval. Thus, the consistency result for our ULDP estimator is not a direct corollary of existing current-status theory but requires a tailored argument.

Before we the main results, we introduce the $L_{p,G}$ consistency with the corresponding $L_{p,G}$ norm.

**Definition 6 ($L_{p,G}$ consistency)** *For a $K$-dimensional function $\mathbf{F}(t)$ and a distribution function $G$, the $L_{p,G}$ norm is $\|\mathbf{F}(t)\|_{p,G}^p = \sum_{k=1}^K \int |F_k(t)|^p dG(t)$. When the distribution $G$ admits a density function $g$ supported on $[0, 1]$, the consistency reduces to the standard $L_p$ consistency with the $L_p$ norm $\|\mathbf{F}(t)\|_p^p = \sum_{k=1}^K \int |F(t)|^p dt$.*

**Theorem 1** *Recalling $T \sim G$ for distribution function $G$ over $[0, 1]$ and when $T$ is independent of $(X, Y)$, one has*

$$\|\mathbf{F}(t) - \widehat{\mathbf{F}}(t)\|_{1,G} = \mathcal{O}_p(\|\mathcal{L}^{-1}\|_{\infty} n^{-1/3}), \|\mathbf{F}(t) - \widehat{\mathbf{F}}(t)\|_{2,G} = \mathcal{O}_p(\lambda_{min}^{-1}(\mathcal{L}) n^{-1/3}),$$

*where $\lambda_{min}(A)$ is the minimum eigenvalue of $A$.*

*Further, if $G$ and $F_{0k}$, $k = 1, \dots, K$ have positive density function $g$ and $f_{0k}$ on $[0, 1]$, then,*

$$\sup_{t \in [0,1]} \|\mathbf{F}(t) - \widehat{\mathbf{F}}(t)\|_{\infty} = \mathcal{O}_p(\|\mathcal{L}^{-1}\|_{\infty} n^{-1/3} \log^{1/3} n).$$

The convergence rates for both $L_p$ and uniform consistency are in line with those of typical shape-constrained estimators, and they also align with the special case when $K = 1$ studied in Liu et al. (2024). For the ACRR algorithm $\lambda_{min}^{-1}(\mathcal{L}) = 1/(1 - e^{-\epsilon})$, which coincides with the reciprocal probability of truthful response of safe inputs.

Next, we establish the point-wise weak convergence result of $\widehat{\mathbf{F}}(t)$, which cannot be improved to simultaneous results on $[0, 1]$ due to nontightness, as explained in Huang & Wellner (1997).

**Theorem 2** *For $t_0 \in (0, 1)$, if $G(t_0)$ and $F_{0k}(t_0), k = 1, \dots, K$, are continuously differentiable at $t_0$ with positive derivatives $g(t_0)$ and $f_{0k}(t_0)$, one has that*

$$n^{1/3}(\mathbf{F}(t_0) - \widehat{\mathbf{F}}(t_0)) \xrightarrow{d} \mathcal{L}^{-1}\mathcal{F}_{t_0}(0),$$

*where the random variable $\mathcal{F}_{t_0}(0)$ is defined in Appendix C.2 due to the space limitations.*

**Remark 5** *Notably, when $K = 1$ and one applies the privacy mechanism in Liu et al. (2024), the point-wise asymptotic distribution $\mathcal{L}^{-1}\mathcal{F}(0)$ will degenerate to*

$$\frac{\left\{4\left(rF_+(t_0) + \frac{1-r}{2}\right)\left(\frac{1+r}{2} - rF_+(t_0)\right)f(t_0)\right\}^{1/3}\arg\max_{t\in\mathbb{R}}\left\{W(t) - t^2\right\}}{(r^2 g(t_0))^{1/3}},$$

*which is consistent with results in Liu et al. (2024). We establish the details in Appendix C.2.*

For the prediction probability over a range of $X$, it is worth noting that for any $0 < t_0 < t_1 < 1$ and $k = 1, \ldots, K$, the conditional probability is given by

$$h_k(t_0, t_1) := \mathbb{P}(Y = k \mid t_0 < X \leqslant t_1) = \frac{\mathbb{P}(Y = k, t_0 < X \leqslant t_1)}{\mathbb{P}(t_0 < X \leqslant t_1)} = \frac{F_{0k}(t_1) - F_{0k}(t_0)}{F_+(t_1) - F_+(t_0)}.$$

Therefore, the conditional probability $h_k(t_0, t_1)$ can be estimated via $\widehat{\mathbf{F}}(t_0)$ and $\widehat{\mathbf{F}}(t_1)$:

$$\widehat{h}_k(t_0, t_1) = \frac{\widehat{F}_{0k}(t_1) - \widehat{F}_{0k}(t_0)}{\widehat{F}_+(t_1) - \widehat{F}_+(t_0)}.$$

The asymptotic properties of $\widehat{h}_k(t_0, t_1)$ follow from Theorems 1 and 2.

**Theorem 3** *Let $0 < t_0 < t_1 < 1$. Suppose that $G(t_0)$, $G(t_1)$, and $F_{0k}(t_0)$, $F_{0k}(t_1)$ for $k = 1, \ldots, K$ are continuously differentiable at $t_0$ and $t_1$ with positive derivatives $g(t_0)$, $g(t_1)$, $f_{0k}(t_0)$, and $f_{0k}(t_1)$, respectively. Then,*

$$\|\mathbf{h}(t_0, t_1) - \widehat{\mathbf{h}}(t_0, t_1)\|_\infty = \mathcal{O}_p\left(\|\mathcal{L}^{-1}\|_\infty n^{-1/3}\log^{1/3} n\right),$$

$$n^{1/3}\left(\mathbf{h}(t_0, t_1) - \widehat{\mathbf{h}}(t_0, t_1)\right) \xrightarrow{d} \frac{\mathcal{L}^{-1}\mathcal{F}_{t_1}(0) - \mathcal{L}^{-1}\mathcal{F}_{t_0}(0)}{\|\mathcal{L}^{-1}\mathcal{F}_{t_1}(0) - \mathcal{L}^{-1}\mathcal{F}_{t_0}(0)\|_+},$$

*where $\mathbf{h}(t_0, t_1) = \{h_k(t_0, t_1)\}_{k=1}^K$, $\widehat{\mathbf{h}}(t_0, t_1) = \{\widehat{h}_k(t_0, t_1)\}_{k=1}^K$, and $\|\mathbf{b}\|_+ := \sum_{k=1}^K b_k$ for a $K$-dimensional vector $\mathbf{b}$.*

At the boundary $t_0 = 0$, we define $\widehat{\mathbf{F}}(t_0) = 0$, ensuring the estimator is well-defined on the full interval $[0, 1]$. The conclusions of Theorem 3 remain valid at $t_0 = 0$. However, at $t_1 = 1$, while consistency still holds, the weak convergence result no longer applies due to the non-differentiable behavior introduced by the ULDP mechanism at the boundary.

**Remark 6** *These boundary cases enable estimation of both $\mathbb{P}(Y = k \mid X \leqslant u)$ and $\mathbb{P}(Y = k \mid X > u)$. The latter is particularly interesting for applications involving censored sensitive regions, as it allows us to estimate the demographic distribution within those zones. However, the estimation quality for $\mathbb{P}(Y = k \mid X > u)$ is generally worse than that for $\mathbb{P}(Y = k \mid X \leqslant u)$, due to reduced label information and the need to estimate $F_{0k}(1)$ (whereas $F_{0k}(0) = 0$ is known by definition).*

## 5 IMPLEMENTATION AND EXPERIMENTS

While theoretical guarantees for our estimator have been established, we introduce several practical implementation strategies that further improve empirical performance.

First, the multi-category case introduces significant computational overhead. Computing the estimator for $n = 10^5$ samples can take about one minute, with runtime growing superlinearly in $n$ (see Appendix E.1). This is consistent with the trend observed in the single-category setting (Liu et al., 2024), but with substantially larger constants. To address this, we adopt a divide-and-conquer strategy: we partition the dataset into $M$ equally sized subsets, compute the estimator on each subset, and average the results. Empirically, we find that setting $M = 4$ both reduces computation time and slightly improves estimation accuracy.

Second, although the estimated CDFs are guaranteed to be non-decreasing and start at zero under the mechanism, randomness and the corrective division by the truthful reporting rate $1 - e^{-\epsilon}$ can cause the total estimated CDF to slightly exceed 1. While prior work such as Liu et al. (2024) proposes

retroactively capping the total CDF at 1, it is not straightforward to apply this constraint to each sub-CDF $F_{0k}$ individually, since the true values $F_{0k}(1)$ (i.e., the marginal category proportions) are unknown. Empirically, we find that capping the total CDF at 1 and stopping the growth of all sub-CDFs beyond that point yields performance very close to an oracle that knows the true marginal proportions. Alternative adjustment rules are compared in Appendix E.2.

For the numerical evaluation, we consider the case $K = 4$, with the true joint CDFs $F_{0k}(x)$ defined as $0.2x$, $0.3x^{1/4}$, $0.3x^4$, and $0.2 \max(0, 3x - 2)$, respectively.

We examine privacy budgets $\epsilon \in \{1, 2, 3\}$, corresponding to strong to moderate privacy regimes. For context, Apple reportedly used $\epsilon = 2$ for sensitive health statistics and $\epsilon = 4$ for emoji usage (Apple, 2020). Sample sizes $n$ range from $10^3$ to $10^6$, with 100 independent replications per setting. To eliminate cross-run correlations, experiments with different sample sizes are conducted independently. As the maximum likelihood estimator is not unique and interpolation may introduce bias, we retain the staircase form of the estimated CDFs for fair comparison.

Performance is evaluated using two metrics: the $L_\infty$ error over $[0, 1]$, and the $\ell_\infty$ error in estimating $\mathbb{P}(Y = k, X < 1/2)$. The means and standard deviations are reported in Table 1. Notably, we can also estimate $\mathbb{P}(Y = k, X > 1/2)$, which corresponds to the distribution within the censored region. The corresponding numerical results are provided in the Appendix E.3.

As shown in Table 2, Appendix E.3, both the uniform consistency and prediction error improve steadily with increasing sample size, confirming the consistency of the estimator. Higher values of $\epsilon$ (weaker privacy) also lead to improved accuracy, as expected. Notably, the prediction error remains reasonably low even under strong privacy constraints ($\epsilon = 1$). We further investigate the relative error in Section E.4 and illustrate our method on real-world data in Section E.5.

## 6 CONCLUSION AND FUTURE WORKS

In this paper, we proposed a flexible ULDP mechanism that adaptively censors potentially sensitive responses without requiring a predefined sensitive region. This is achieved through (i) transforming and privatizing binary sensitivity indicators and (ii) applying a randomized-response step. Our two-stage design decouples privacy preservation from statistical estimation: the privatized data can be interpreted as truthful samples from a transformed variable, and the CDF estimator is computed via a bound-constrained discretized maximum-likelihood procedure.

We establish that the proposed estimator achieves $L^2$- and sup-norm consistency at rates $\mathcal{O}_p(n^{-1/3})$ and $\mathcal{O}_p(n^{-1/3} \log n)$, respectively, with pointwise weak convergence in the interior following the classical Chernoff limit distribution. These theoretical guarantees extend naturally to multi-category prediction under ULDP. Simulations confirm both the practical accuracy and computational viability of the approach across a range of settings, marking the first rigorous treatment of ULDP-based CDF estimation and prediction with provable asymptotic properties.

Despite these contributions, several limitations remain. First, the reconstruction of the distribution is based on a capping mechanism, which enforces a fixed upper bound on the estimated CDF. While effective, this approach may obscure the true distributional structure near the boundary. A potentially more informative direction would involve jointly estimating the CDF under the constraint, possibly leading to richer theoretical insights.

Although the proposed method is computationally efficient for small to moderate numbers of categories, the complexity increases substantially with $K$. In particular, the runtime grows significantly compared to the $K = 1$ case, making the approach less practical for applications involving very large datasets (e.g., $n > 10^8$). Developing scalable algorithms or approximation techniques for high-throughput scenarios is thus a valuable avenue for future research.

Finally, like other nonparametric estimators under privacy constraints, it does not achieve the parametric $\mathcal{O}_p(n^{-1/2})$ rate of the non-private empirical CDF. There are some potential approaches to improve the cube-root rate estimator, such as smoothed estimator (Groeneboom et al., 2010) and the federated estimator (Shi et al., 2018). We observe similar improvement via the later approach in Section E.1. Addressing these statistical limitations, while preserving ULDP guarantees, remains an important and challenging open problem.

REPRODUCIBILITY STATEMENT

All numerical experiments and real-data analyses are fully reproducible via the code included in the submitted anonymized supplementary materials.

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

## A   A SAMPLE SURVEY BASED ON ACRR

Assuming that researchers aim to estimate how outstanding credit-card debt ($X$) is distributed across universities ($Y$) while safeguarding respondents with high debt.

Disclosing a *high* debt level together with an identifying attribute (alma mater) raises privacy concerns. We therefore treat "high debt" as the sensitive direction and design a survey that never reveals both high debt and university simultaneously.

**Survey protocol for each participant**

1. Sample a personal threshold $T \sim G$ (for example, $G$ uniform on $[0, \$25\,000]$).

2. Ask the participant to follow the random procedure below.

**Credit-Card Debt and Alma Mater Questionnaire**

1. **Your threshold:** $T = \$18\,000$.

2. **Flip a fair coin.**
   Heads: mark **Yes** in Step 4 and skip Step 3.
   Tails: continue to Step 3.

3. Compare your own debt $x$ with $T$.
   If $x > T$: mark **Yes**.
   If $x \leqslant T$: tick the university you graduated from.

4. **Select exactly one option**
   ▫ **Yes**
   ▫ University A      ▫ University B      ▫ University C

This protocol offers plausible deniability that some low-debt participants output "Yes" solely because the coin landed heads, so high-debt participants who also answer "Yes" are $\epsilon$ indistinguishable from them ($\epsilon \approx 0.7$). Conditional disclosure ensures that only low-debt respondents with a tails outcome and $x \leqslant T$ reveal their university. There is no joint leakage because the final response space contains either "Yes" or one university label—never both.

Despite the heavy censoring of individual responses, the underlying distribution can still be accurately reconstructed by applying the recovery procedure described in Section 3.3.

# B  FULL MECHANISM OF ACRR

---

**Algorithm 2** Perturbation (client-side ACRR reporting)

**Inputs:**

- Privacy parameter $\epsilon > 0$.
- Threshold distribution $G$ on $[0, 1]$ (public).
- User's private pair $(x_i, y_i) \in [0, 1] \times \{1, \ldots, K\}$.

**Output:** A one–hot report $a_i \in \{e_1, \ldots, e_K, e_{K+1}\}$ and the issued threshold $t_i$.

**Procedure:**

1. Draw $t_i \sim G$ (issued to user $i$).
2. Compute $b_i \leftarrow \mathbf{1}\{x_i > t_i\}$.
3. Apply ACRR:
   - If $b_i = 1$, set $a_i \leftarrow e_{K+1}$.
   - If $b_i = 0$, set

$$a_i \leftarrow \begin{cases} e_{y_i}, & \text{with probability } 1 - e^{-\epsilon}, \\ e_{K+1}, & \text{with probability } e^{-\epsilon}. \end{cases}$$

4. Send $(t_i, a_i)$ to the aggregator.

---

## C  PROOF OF MAIN RESULTS

### C.1  PROOF OF THEOREM 1

Since $\mathbb{P}(\Delta_i|X_i, Y_i, T_i) = P(\Delta^\star|X_i, Y_i, T_i)$, one can transform the CDF $\mathbf{F}$ estimation with data $\Delta_i$ under LDP into CDF $\mathbf{F}^\star$ estimation with data $\Delta_i^\star$ under non-DP. Using $\Delta_i^\star$ to recover $\mathbf{F}^\star$ is a typical current status problem. Applying Theorem 4.1 and Collary 4.2 in Groeneboom et al. (2008a), one obtains the $L_{1,G}$ and $L_{2,G}$ consistency with order $n^{-1/3}$,i.e.,

$$\|\mathbf{F}^\star(t) - \widehat{\mathbf{F}}^\star(t)\|_{1,G} = \mathcal{O}_p(n^{-1/3}), \|\mathbf{F}^\star(t) - \widehat{\mathbf{F}}(t)\|_{2,G} = \mathcal{O}_p(n^{-1/3}).$$

Based on the linear mapping between $(\mathbf{F}^\star, \widehat{\mathbf{F}}^\star)$ and $(\mathbf{F}, \widehat{\mathbf{F}})$, one has that

$$\|\mathbf{F}(t) - \widehat{\mathbf{F}}(t)\|_{1,G} = \mathcal{O}_p(\|\mathcal{L}^{-1}\|_\infty n^{-1/3}), \|\mathbf{F}(t) - \widehat{\mathbf{F}}(t)\|_{2,G} = \mathcal{O}_p(\lambda_{min}^{-1}(\mathcal{L})n^{-1/3}).$$

For uniform consistency, due to the CDF function $\mathbf{F}^\star$ is not absolutely continuous, it is not trivial to apply the existing results of current status problem. We will derive the local consistency on some interval first, and then combines this intervals to establishes the uniform consistency.

In detail, for $t_0 \in [0, 1]$, one denotes the interval $\mathcal{I}_\omega(t_0) = (t_0 - \omega, t_0 + \omega)$, if $F_+^\star(t_0) > 0$, $\mathcal{I}_r(t_0) = (t_0, t_0 + \omega)$ if $F_+^\star(t_0) = 0$, and $\mathcal{I}_r(t_0) = (t_0 - \omega, t_0)$ if $F_+^\star(t_0) = 1$, for some $\omega > 0$. Notices that if $F_{0k}, k = 1, \ldots, K$ have positive density function $g$ and $f_{0k}$ on $[0, \gamma]$, then $F_{0k}^\star$'s are continuously differentiable at $t_0$ with positive and bounded away from zero derivatives in interval $\mathcal{I}_r(t_0)$ for any $t_0 \in [0, 1]$ and some $\omega > 0$. Therefore, according to Lemmas 4.1 and 4.4 of Malov (2021), one has that

$$\sup_{t \in \mathcal{I}_\omega(t_0)} \|\mathbf{F}^\star(t) - \widehat{\mathbf{F}}^\star(t)\|_\infty = \mathcal{O}_p(n^{-1/3} \log^{1/3} n).$$

Recalling that $[0, 1]$ is a compact set, we select a finite cover $\{\mathcal{I}_{\omega_j}(t_j)\}_{j=1}^d$ of the interval. Then we find that

$$\sup_{t \in [0,1]} \|\mathbf{F}^\star(t) - \widehat{\mathbf{F}}^\star(t)\|_\infty = \max_{j \in \{1,\ldots,d\}} \sup_{t \in \mathcal{I}_{\omega_j}(t_j)} \|\mathbf{F}^\star(t) - \widehat{\mathbf{F}}^\star(t)\|_\infty = \mathcal{O}_p(n^{-1/3} \log^{1/3} n).$$

Finally, Based on the linear mapping between $(\mathbf{F}^\star, \widehat{\mathbf{F}}^\star)$ and $(\mathbf{F}, \widehat{\mathbf{F}})$, one has that

$$\sup_{t \in [0,1]} \|\mathbf{F}(t) - \widehat{\mathbf{F}}(t)\|_\infty = \mathcal{O}_p(\|\mathcal{L}^{-1}\|_\infty n^{-1/3} \log^{1/3} n).$$

### C.2  PROOF OF THEOREM 2

To introduce our pointwise asymptotic results, we first define the distribution $\mathcal{F}_{t_0}$

Let $\mathbf{W} = (W_1, \ldots, W_K)$ be a $K$-tuple of two-sided Brownian motion processes originating from zero, with mean zero and covariances

$$E\{W_j(t)W_k(s)\} = (|s| \wedge |t|)1\{st > 0\}\Sigma_{jk}, \quad s, t \in \mathbb{R}, j, k \in \{1, \ldots, K\},$$

where

$$\Sigma_{jk} = g(t_0)^{-1}\{1\{j = k\}F_{0k}^\star(t_0) - F_{0j}^\star(t_0)F_{0k}^\star(t_0)\}.$$

Moreover, $\mathbf{V}_{t_0} = (V_{1,t_0}, \ldots, V_{K,t_0})$ is a vector of drifted Brownian motions, defined by

$$V_{k,t_0}(t) = W_k(t) + \frac{1}{2}f_{0k}^\star(t_0)t^2, \quad k = 1, \ldots, K$$

Similarly, let $V_{+,t_0} = \sum_{k=1}^K V_{k,t_0}, W_+ = \sum_{k=1}^K W_k$. Following Theorem 1.7 in Groeneboom et al. (2008b), for some $t_0 \in (0, 1)$, there exists an almost surely unique $K$-tuple $\widehat{\mathbf{H}}_{t_0} = (\widehat{H}_{1,t_0}, \ldots, \widehat{H}_{K,t_0})$ of convex functions with right-continuous derivatives $\mathcal{F}_{t_0}(t) = (\mathcal{F}_{1,t_0}(t), \ldots, \mathcal{F}_{K,t_0}(t))$ satisfying the following three conditions, where $a_{k,t_0} = (F_{0k}(t_0))^{-1}$, and $a_{K+1,t_0} = (1 - F_+(t_0))^{-1}$,

- $a_{k,t_0}\widehat{H}_{k,t_0}(t) + a_{K+1,t_0}\widehat{H}_{+,t_0}(t) \leqslant a_{k,t_0}V_{k,t_0}(t) + a_{K+1,t_0}V_{+,t_0}(t)$, for $k = 1,\ldots,K, t \in \mathbb{R}$.

- $\int \left\{ a_{k,t_0}\widehat{H}_{k,t_0}(t) + a_{K+1,t_0}\widehat{H}_{+,t_0}(t) - a_{k,t_0}V_{k,t_0}(t) - a_{K+1,t_0}V_{+,t_0}(t) \right\} d\widehat{F}_k(t) = 0, k = 1,\ldots,K$.

- For all $M > 0$ and $k = 1,\ldots,K$, there are points $\tau_{1k} < -M$ and $\tau_{2k} > M$ so that $a_k\widehat{H}_{k,t_0}(t) + a_{K+1,t_0}\widehat{H}_{+,t_0}(t) = a_{k,t_0}V_{k,t_0}(t) + a_{K+1,t_0}V_{+,t_0}(t)$ for $t = \tau_{1k}$ and $t = \tau_{2k}$.

Similarly, for $t_0 \in (0,1)$, if $F_{0k}(t_0), k = 1,\ldots,K$, are continuously differentiable at $t_0$ with positive derivatives $f_{0k}(t_0)$, then $F_{0k}^\star(t_0), k = 1,\ldots,K$, are continuously differentiable at $t_0$ with positive derivatives $f_{0k}^\star(t_0)$. One applies Theorem 1.8 of Groeneboom et al. (2008b), and obtains

$$n^{1/3}(\mathbf{F}^\star(t_0) - \widehat{\mathbf{F}}^\star(t_0)) \xrightarrow{d} \mathcal{F}_{t_0}(0).$$

Combined with continuous mapping theorem, the proof is completed.

If $K = 1$ and one applies the privacy mechanism in Liu et al. (2024), then one only needs to estimate $F_+(t)$. Then relationship $\mathbf{F}^\star(t) = \mathcal{L}\mathbf{F}(t)\mathbf{1}_{0<t<1} + \mathbf{1}_{t=1}$ will degenerate to $F_+^\star(t) = \{rF_+(t) + (1-r)/2\}_{0<t<1} + \mathbf{1}_{t=1}$ and $f_+^\star(t) = rf(t)$. Hence,, the variance term $\Sigma_{jk}$ of two-sided Brownian motion $W$ will degenerate to

$$\frac{\left\{ 4\left(rF_+(t_0) + \frac{1-r}{2}\right)\left(\frac{1+r}{2} - rF_+(t_0)\right) \right\}}{(g(t_0))}.$$

Based on the relationships between Brownian motion and Chernoff distribution, see Groeneboom (1989),

$$\arg\max_{t\in\mathbb{R}} \left\{ W(t) - ct^2 \right\} \stackrel{d}{=} c^{-1/3}\mathcal{F}_{t_0}(0) \stackrel{d}{=} c^{-1/3}\arg\max_{t\in\mathbb{R}} \left\{ W(t) - t^2 \right\},$$

for some $c > 0$. Let $c = \frac{\left\{ 4\left(rF_+(t_0) + \frac{1-r}{2}\right)\left(\frac{1+r}{2} - rF_+(t_0)\right)f(t_0) \right\}}{(g(t_0))}$, the point-wise asymptotic distribution $\mathcal{L}^{-1}\mathcal{F}(0)$ will degenerate to

$$\frac{\left\{ 4\left(rF_+(t_0) + \frac{1-r}{2}\right)\left(\frac{1+r}{2} - rF_+(t_0)\right)f(t_0) \right\}^{1/3}\left(\arg\max_{t\in\mathbb{R}}\left\{ W(t) - t^2 \right\},\right)}{(r^2 g(t_0))^{1/3}}.$$

### C.3 PROOF OF THEOREM 3

The consistent result is derived by Theorem 1 deriectly.

For second result, one notice that for any $0 < t_0 < t_1 < 1$, $\mathbf{F}(t_0) - \widehat{\mathbf{F}}(t_0)$ and $\mathbf{F}(t_1) - \widehat{\mathbf{F}}(t_1)$ are asymptotically independent, see page 131 in Huang & Wellner (1997) for the local dependence structure of this type of process in a closely related problem. Then, following Theorem 2,

$$n^{1/3}\left\{ \mathbf{F}(t_0) - \widehat{\mathbf{F}}(t_0)), \mathbf{F}(t_1) - \widehat{\mathbf{F}}(t_1)) \right\} \xrightarrow{d} \left\{ \mathcal{F}_{t0}(0), \mathcal{F}_{t1}(0) \right\}.$$

Apply the continuous mapping theorem, the theorem is proved.

## D  ICM ALGORITHM

In this section, we briefly introduce the ICM algorithm following Groeneboom & Jongbloed (2014). The notation remains consistent with that used above.

Denote by $e_k$ the $k$-th unit vector in $\mathbb{R}^K$, by $\#$ the counting measure on $D = \{e_k : k = 1,\ldots,K+1\}$, and by $G$ the distribution of $T$. We define the measure $\mu = G \times \#$ on $\mathbb{R} \times D$. With respect to this dominating measure, the density of a single observation $(T, \Delta)$ is

$$p_F(t, \delta) = \prod_{k=1}^{K} F_k(t)^{\delta_k}\left(1 - F_+(t)\right)^{1-\delta_+}, \tag{3}$$

where $\delta_+ = \sum_{k=1}^{K} \delta_k$. Given an independent sample of size $n$ distributed as $(T, \Delta)$,

$$(t_i, \Delta_i) = (t_i, \Delta_{i,1}, \ldots, \Delta_{i,K+1}), \quad i = 1, \ldots, n,$$

the log-likelihood 1 is

$$\ell(F) = \int \log p_F(t, \delta) \, d\mathbb{P}_n(t, \delta)$$

$$= \int \left\{ \sum_{k=1}^{K} \delta_k \log F_k(t) + (1 - \delta_+) \log(1 - F_+(t)) \right\} d\mathbb{P}_n(t, \delta),$$

where $\mathbb{P}_n$ is the empirical distribution of $(t_i, \Delta_i)$. An MLE $\hat{F} = (\hat{F}_1, \ldots, \hat{F}_K)$ is defined as

$$\ell(\hat{F}) = \max_{F \in \mathcal{F}_K} \ell(F),$$

where

$$\mathcal{F}_K = \big\{ F = (F_1, \ldots, F_K) : F_1, \ldots, F_K \text{ are subdistribution functions,}$$

$$\text{and for all } x \geqslant 0, \ \sum_{k=1}^{K} F_k(x) \leqslant 1 \big\}.$$

The iterative convex minorant (ICM) algorithm is derived from Corollary 2.10 in Groeneboom et al. (2008a), restated below.

**Lemma 1 (Corollary 2.10 in Groeneboom et al. (2008a))** *Let*

$$\lambda = 1 - \int \frac{\delta_{K+1}}{1 - \hat{F}_+(u)} \, d\mathbb{P}_n(u, \delta) \geqslant 0. \tag{4}$$

*Then $\hat{F} = (\hat{F}_1, \ldots, \hat{F}_K)$ is an MLE if, for all $k = 1, \ldots, K$ and each point of jump $\tau_{ki}$ of $\hat{F}_k$,*

$$\int_{u \in [\tau_{ki}, s)} \left\{ \frac{\delta_k}{\hat{F}_k(u)} - \frac{\delta_{K+1}}{1 - \hat{F}_+(u)} \right\} d\mathbb{P}_n(u, \delta) \geqslant \lambda \mathbb{1}_{[\tau_{ki}, s)}(T_{(p)}), \quad s \in \mathbb{R}, \tag{5}$$

*where equality holds if $s > \tau_{ki}$ is a point of increase of $\hat{F}_k$, or if $s > T_{(p)}$, the largest strictly ordered order statistic.*

A cusum diagram is then constructed with points $(0, 0)$ and

$$\left( \sum_{i=1}^{j} w_{ki}, \ \sum_{i=1}^{j} (w_{ki} y_{ki} + v_{ki}) - \lambda \mathbb{1}_{\{j = n_k\}} \right), \quad j = 1, \ldots, n_k, \tag{6}$$

where $\lambda$ is defined in equation 4 using the current iterate $F_+$,

$$w_{ki} = \int_{u \in [T_i, T_j)} \left\{ \frac{\delta_k}{F_k(u)^2} + \frac{\delta_{K+1}}{\left(1 - F_+(u)\right)^2} \right\} d\mathbb{P}_n(u, \delta),$$

with $T_i$ and $T_j$ successive points where $\delta_k = 1$,

$$v_{ki} = \int_{u \in [T_i, T_j)} \left\{ \frac{\delta_k}{F_k(u)} - \frac{\delta_{K+1}}{1 - F_+(u)} \right\} d\mathbb{P}_n(u, \delta),$$

and $y_{ki}$ is the value of $F_k$ at $T_i$ in the current iteration. The quantity $n_k$ denotes the number of points where $\delta_k = 1$. As described in Groeneboom & Jongbloed (2014), the $w_{ki}$ correspond to the diagonal elements of the Hessian of the maximization problem.

The next step is to compute the greatest convex minorant of the cusum diagram equation 6 and use its left derivative to update $F_k$. The Lagrange multiplier $\lambda$ is recomputed in each iteration using equation 4. We only need to update $F_k$ at points where $\delta_k = 1$, since these are the only locations where the subdistribution can place mass. At any stationary point of the iteration, the conditions in equation 5 are satisfied, and hence an MLE is obtained. A golden section search is used to determine the step size for each iteration.

# E  ADDITIONAL NUMERICAL RESULTS

## E.1  DIVIDE AND CONQUER IN THE ICM ALGORITHM

In this subsection, we are using the same setting as in Chapter 5, where we set $\epsilon = 1$. All experiments are run on a single core of an AMD 9950X CPU. The figure below illustrates the mean and median computational time when running the ICM algorithm:

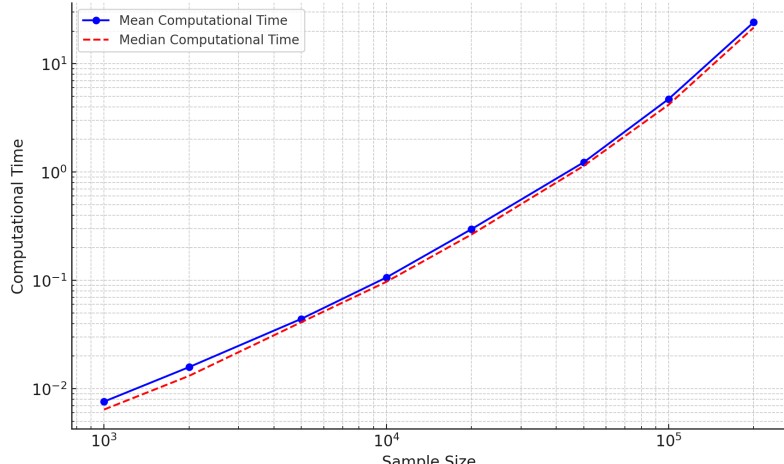

Figure 2: Mean and median computational time for the ICM algorithm

The plot demonstrates that it takes approximately one second to process $n = 50000$ data points. Linear regression shows that computational time increases at a rate of approximately $n^{1.49}$.

Since this rate is superlinear, we consider a divide-and-conquer strategy, where the data is randomly split into four even-sized portions, and the final result is summarized by taking the average.

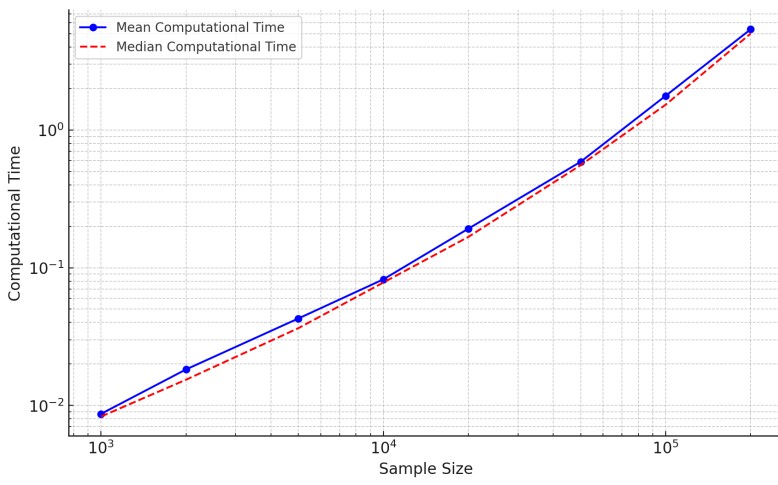

Figure 3: Computational time under the divide-and-conquer strategy

Under this strategy, the growth rate of computational time is significantly reduced. However, for smaller sample sizes, the overhead associated with splitting the data may result in slightly longer runtimes. However, the total computational time for small datasets remains under one second.

Importantly, improvement in computational efficiency does not come at the cost of accuracy of estimation. In contrast, all divide-and-conquer strategies we considered, namely splitting the data

into 2, 4, and 8 parts, consistently produce better results compared to the baseline (without division). We illustrate this below for the case of $\epsilon = 1$.

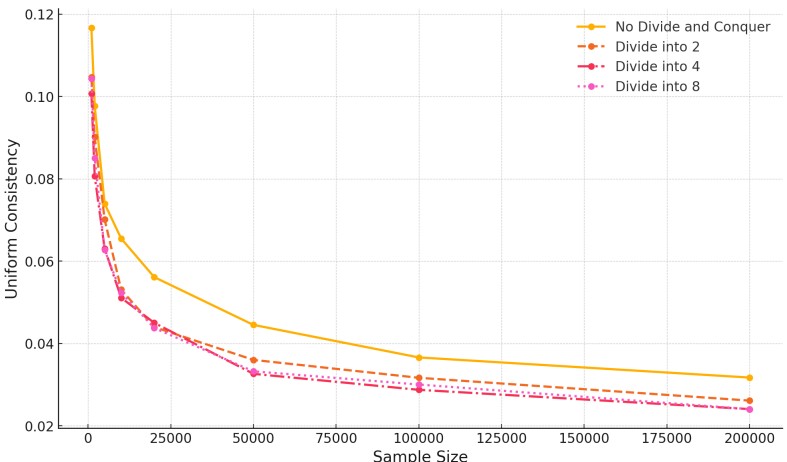

Figure 4: Error Under Different Divide-and-Conquer Settings

The observed improvement may be partly explained by Theorems 1 and 2, which show that the variance of the proposed ULDP estimator is of order $n^{1/3}$. When the data set is divided into subsets $L$, the variance of the resulting divide-and-conquer estimator is expected to scale heuristically as $L^{-1/6}n^{1/3}$. However, the impact of bias under this strategy remains unclear and is analytically difficult to characterize. In particular, in the extreme case of $L = n$, the estimator reduces to the empirical cumulative distribution function, which almost surely converges to an incorrect distribution. This suggests the existence of an optimal choice of $L$, although identifying it theoretically is a complex problem beyond the scope of this paper.

## E.2 COMPARING TO ALTERNATIVE ORACLE CAPPING MECHANISM

Although having the estimated CDF exceed 1 does not violate the theoretical error bounds presented in our main results, it can be problematic in practical, real-world streaming applications. Formally, the aggregated estimated CDF $\hat{F}_0(t) = \sum_{k=1}^{K} \hat{F}_{0k}(t)$ can sometimes surpass 1 due to randomness and adjustments from the ULDP mechanism. To address this, we propose a method that ensures the estimates remain interpretable and potentially improve accuracy.

We introduce a correction based on the rule of stopping at 1, denoted as $\check{F}_{0k}^{stop\_at\_one}(t)$:

$$\check{F}_{0k}^{\text{stop\_at\_one}}(t) = \begin{cases} \hat{F}_{0k}(t), & \text{if } \hat{F}_0(t) \leqslant 1, \\ \lim_{s \to \inf\{u>0:\hat{F}_0(u)>1\}^-} \hat{F}_{0k}(s), & \text{otherwise.} \end{cases}$$

This method halts the growth of all sub-CDF estimates simultaneously at the earliest point where the aggregate estimate first exceeds unity, thereby maintaining monotonicity and interpretability without altering previous estimates.

To evaluate its performance, we compare against an oracle based method, denoted $\check{F}_{0k}^{oracle}$. This oracle assumes knowledge of the true marginal values $F_{0k}(1)$, and thus caps each sub-CDF estimate at the true marginal proportion:

$$\check{F}_{0k}^{oracle}(t) = \min(\hat{F}_{0k}(t), F_{0k}(1)),$$

Although practically unattainable, this oracle serves as an optimal performance benchmark. We find that by applying the stopping rule, the corrected estimates can reach results similar to the oracle, offering a practical solution that ensures both interpretability and accuracy in streaming applications.

The empirical results comparing these methods are presented below.

Table 1: Empirical results of uniform consistency (standard deviation) under the proposed capping mechanism and the oracle method

| $n$ | Stop at one | | | Oracle | | |
|---|---|---|---|---|---|---|
| | $\epsilon = 1$ | $\epsilon = 2$ | $\epsilon = 3$ | $\epsilon = 1$ | $\epsilon = 2$ | $\epsilon = 3$ |
| $1 \times 10^3$ | 0.098(0.024) | 0.085(0.018) | 0.086(0.019) | 0.082(0.015) | 0.075(0.016) | 0.074(0.014) |
| $2 \times 10^3$ | 0.082(0.020) | 0.071(0.018) | 0.066(0.016) | 0.072(0.016) | 0.063(0.013) | 0.059(0.014) |
| $5 \times 10^3$ | 0.062(0.013) | 0.055(0.012) | 0.054(0.010) | 0.055(0.011) | 0.051(0.011) | 0.049(0.013) |
| $1 \times 10^4$ | 0.051(0.012) | 0.045(0.010) | 0.044(0.010) | 0.045(0.009) | 0.042(0.008) | 0.041(0.009) |
| $2 \times 10^4$ | 0.041(0.010) | 0.038(0.008) | 0.037(0.009) | 0.039(0.009) | 0.037(0.008) | 0.036(0.009) |
| $5 \times 10^4$ | 0.034(0.008) | 0.029(0.006) | 0.028(0.006) | 0.031(0.007) | 0.030(0.007) | 0.029(0.007) |
| $1 \times 10^5$ | 0.029(0.006) | 0.025(0.005) | 0.024(0.006) | 0.027(0.007) | 0.025(0.006) | 0.024(0.005) |
| $2 \times 10^5$ | 0.024(0.005) | 0.022(0.005) | 0.021(0.005) | 0.024(0.006) | 0.021(0.005) | 0.021(0.005) |
| $5 \times 10^5$ | 0.019(0.005) | 0.018(0.004) | 0.017(0.005) | 0.019(0.004) | 0.017(0.003) | 0.017(0.003) |
| $1 \times 10^6$ | 0.017(0.004) | 0.014(0.004) | 0.013(0.003) | 0.016(0.003) | 0.014(0.003) | 0.013(0.003) |

Notably, the proposed capping mechanism achieves performance close to the oracle, particularly for large sample sizes; therefore, we recommend using this correction in conjunction with our method, and it is adopted in all empirical evaluations presented in this paper.

## E.3 TABLES IN SECTION 5

Table 2: Empirical results of uniform consistency and prediction error (standard deviation))

| $n$ | Uniform Consistency | | | Prediction Error | | |
|---|---|---|---|---|---|---|
| | $\epsilon = 1$ | $\epsilon = 2$ | $\epsilon = 3$ | $\epsilon = 1$ | $\epsilon = 2$ | $\epsilon = 3$ |
| $1 \times 10^3$ | 0.098(0.024) | 0.085(0.018) | 0.086(0.019) | 0.059(0.036) | 0.053(0.029) | 0.053(0.028) |
| $2 \times 10^3$ | 0.082(0.020) | 0.071(0.018) | 0.066(0.016) | 0.042(0.025) | 0.041(0.020) | 0.036(0.019) |
| $5 \times 10^3$ | 0.062(0.013) | 0.055(0.012) | 0.054(0.010) | 0.030(0.016) | 0.028(0.014) | 0.028(0.015) |
| $1 \times 10^4$ | 0.051(0.012) | 0.045(0.010) | 0.044(0.010) | 0.023(0.014) | 0.020(0.013) | 0.019(0.010) |
| $2 \times 10^4$ | 0.041(0.010) | 0.038(0.008) | 0.037(0.009) | 0.019(0.010) | 0.018(0.010) | 0.016(0.009) |
| $5 \times 10^4$ | 0.034(0.008) | 0.029(0.006) | 0.028(0.006) | 0.013(0.007) | 0.011(0.005) | 0.010(0.006) |
| $1 \times 10^5$ | 0.029(0.006) | 0.025(0.005) | 0.024(0.006) | 0.011(0.006) | 0.009(0.005) | 0.009(0.005) |
| $2 \times 10^5$ | 0.024(0.005) | 0.022(0.005) | 0.021(0.005) | 0.008(0.004) | 0.007(0.004) | 0.007(0.004) |
| $5 \times 10^5$ | 0.019(0.005) | 0.018(0.004) | 0.017(0.005) | 0.006(0.004) | 0.005(0.003) | 0.005(0.003) |
| $1 \times 10^6$ | 0.017(0.004) | 0.014(0.004) | 0.013(0.003) | 0.005(0.002) | 0.004(0.002) | 0.003(0.002) |

Table 3: Prediction error: mean (standard deviation) of $\mathbb{P}(Y = k, X > 1/2)$.

| $n$ | Prediction Error | | |
|---|---|---|---|
| | $\epsilon = 1$ | $\epsilon = 2$ | $\epsilon = 3$ |
| $1 \times 10^3$ | 0.138 (0.055) | 0.108 (0.046) | 0.107 (0.045) |
| $2 \times 10^3$ | 0.114 (0.042) | 0.090 (0.036) | 0.087 (0.034) |
| $5 \times 10^3$ | 0.080 (0.029) | 0.073 (0.025) | 0.064 (0.028) |
| $1 \times 10^4$ | 0.067 (0.024) | 0.058 (0.023) | 0.054 (0.025) |
| $2 \times 10^4$ | 0.054 (0.021) | 0.046 (0.016) | 0.041 (0.017) |
| $5 \times 10^4$ | 0.037 (0.015) | 0.034 (0.015) | 0.034 (0.013) |
| $1 \times 10^5$ | 0.031 (0.012) | 0.028 (0.011) | 0.024 (0.011) |
| $2 \times 10^5$ | 0.025 (0.009) | 0.020 (0.009) | 0.019 (0.009) |
| $5 \times 10^5$ | 0.017 (0.007) | 0.015 (0.006) | 0.014 (0.005) |
| $1 \times 10^6$ | 0.013 (0.004) | 0.011 (0.004) | 0.009 (0.004) |

### E.4 RELATIVE ERROR ANALYSIS

In addition to the uniform consistency in Section 5, we also consider relative uniform consistency, defined as

$$\sup_{t \in (0,1)} \max_{k \in \{1,2,3,4\}} \frac{F_{0k}(t) - \widehat{F}_k(t)}{F_{0k}(1)}.$$

The numerical results are provided in the table at the end of this response. Since $F_{0k}(1) \in [0.2, 0.3]$, we expect the relative error to be inflated by a factor between $1/0.3 \approx 3.33$ and $1/0.2 = 5$. Empirically, the observed inflation factor is about $3.74$ on average with standard deviation $0.23$, which lies within the theoretical range. This indicates that the maximum relative error is not concentrated only on the most frequent or the least frequent categories.

Furthermore, we do not observe any significant differences in the relative error across different values of $\epsilon$. For sufficiently large $n$, the relative errors remain reasonable, implying no notable increase in error for less common categories.

Table 4: Relative Empirical results of uniform consistency and prediction error (standard deviation).

| $n$ | Uniform consistency | | |
| --- | --- | --- | --- |
| | $\epsilon = 1$ | $\epsilon = 2$ | $\epsilon = 3$ |
| $1 \times 10^3$ | 0.416 (0.110) | 0.348 (0.079) | 0.337 (0.069) |
| $2 \times 10^3$ | 0.335 (0.075) | 0.283 (0.060) | 0.263 (0.056) |
| $5 \times 10^3$ | 0.233 (0.053) | 0.210 (0.044) | 0.207 (0.042) |
| $1 \times 10^4$ | 0.190 (0.037) | 0.176 (0.040) | 0.160 (0.035) |
| $2 \times 10^4$ | 0.161 (0.034) | 0.144 (0.031) | 0.132 (0.024) |
| $5 \times 10^4$ | 0.125 (0.025) | 0.113 (0.025) | 0.106 (0.020) |
| $1 \times 10^5$ | 0.102 (0.020) | 0.088 (0.016) | 0.084 (0.014) |
| $2 \times 10^5$ | 0.083 (0.017) | 0.075 (0.017) | 0.073 (0.016) |
| $5 \times 10^5$ | 0.068 (0.016) | 0.061 (0.014) | 0.060 (0.013) |
| $1 \times 10^6$ | 0.061 (0.015) | 0.054 (0.013) | 0.051 (0.013) |

### E.5 REAL DATA ANALYSIS

For validation of the proposed method on real-world data, we used the government salary dataset available in the R package `fairadapt`, which contains 204,309 salary records. In this dataset, salary is treated as the continuous response and race (7 categories) as the categorical variable. Although this dataset is not privacy-sensitive in the sense of containing ground-truth protected attributes for disclosure, it serves as a practical validation example.

We applied the following preprocessing steps. We removed outliers with salaries exceeding $200,000, which account for less than 0.2% of the records. Following the same approach as for the synthetic data, we randomly sampled without replacement subsets of sizes 5,000, 10,000, 20,000, 50,000, 100,000, and 200,000 (nearly the full dataset). Smaller sample sizes (e.g., 1,000 and 2,000) were excluded because some race categories would be absent. For each sample size, we ran experiments with privacy budgets $\epsilon \in \{1, 2, 3\}$. Each setting was repeated for 100 independent repetitions. The reported quantities are the average $L_\infty$ errors with standard deviations (in parentheses).

Table 5: Consistency on real data

| $n$ | $\epsilon$ | | |
|---|---|---|---|
| | 1 | 2 | 3 |
| $5 \times 10^3$ | 0.084 (0.019) | 0.075 (0.017) | 0.073 (0.016) |
| $1 \times 10^4$ | 0.067 (0.013) | 0.060 (0.012) | 0.058 (0.011) |
| $2 \times 10^4$ | 0.058 (0.012) | 0.049 (0.009) | 0.044 (0.008) |
| $5 \times 10^4$ | 0.044 (0.008) | 0.040 (0.007) | 0.039 (0.007) |
| $1 \times 10^5$ | 0.038 (0.006) | 0.035 (0.007) | 0.033 (0.006) |
| $2 \times 10^5$ | 0.033 (0.005) | 0.029 (0.005) | 0.028 (0.004) |

These results demonstrate performance comparable to the synthetic data, with a slightly higher error primarily attributable to class imbalance: approximately $90\%$ of observations belong to the White group.

# F    THE VARIANTS OF ACRR

The ACRR mechanism can be modified to conform with the classical definition of local differential privacy. Consider the following randomized mechanism $\mathcal{M}_{\text{LDP1}} : X \to \mathcal{E}$, which satisfies $\epsilon$-LDP:

$$
\mathcal{M}_{\text{LDP1}}((0,k)) = \begin{cases} e_j & \text{with probability } \frac{1}{K+e^\epsilon} \text{ for } j \neq k, \\ e_k & \text{with probability } \frac{e^\epsilon}{K+e^\epsilon}. \end{cases}
$$

$$
\mathcal{M}_{\text{LDP1}}((1,k)) = \begin{cases} e_j & \text{with probability } \frac{1}{K+e^\epsilon} \text{ for } j = 1, \cdots, K, \\ e_{K+1} & \text{with probability } \frac{e^\epsilon}{K+e^\epsilon}. \end{cases}
$$

This guarantees $\epsilon$-LDP, as the ratio of any two output probabilities is bounded by $e^\epsilon$.

Although this mechanism introduces more noise compared to ACRR, it can still be analyzed using the framework using the framework developed in Chapter 4. In particular, the associated perturbation matrix $\mathcal{L} \in \mathbb{R}^{(K+1) \times (K+1)}$ has the following structure:

$$
\mathcal{L}_{LDP1} = \frac{1}{K+e^\epsilon} \begin{bmatrix} e^\epsilon & 1 & \cdots & 1 \\ 1 & e^\epsilon & \cdots & 1 \\ \vdots & \vdots & \ddots & \vdots \\ 1 & 1 & \cdots & e^\epsilon \end{bmatrix}.
$$

Notice that according to Theorems 1, 2, and 3, the matrix $\mathcal{L}_{LDP1}^{-1}$ acts as a multiplicative factor in the error bound. A direct computation shows that:

$$
\|\mathcal{L}_{LDP1}^{-1}\|_\infty = \frac{1 + e^{-\epsilon}(2K-1)}{1 - e^{-\epsilon}},
$$

whereas for the ACRR mechanism, we have:

$$
\|\mathcal{L}^{-1}\|_\infty = \frac{1 + e^{-\epsilon}}{1 - e^{-\epsilon}}.
$$

This indicates that *asymmetric privacy protection* eliminates the inflation dependent on $K$ in the error bound.

Even though LDP1 performs poorly in other respects, it still benefits from the censoring mechanism. To see why, consider a standard LDP2 random response mechanism that perturbs $X$ to $\tilde{X}$ (rather than to $\mathcal{E}$). In this case, the perturbation matrix becomes a $2K \times 2K$ matrix:

$$
\mathcal{L}_{\text{LDP2}} = \frac{1}{2K + e^\epsilon - 1} \begin{bmatrix} e^\epsilon & 1 & \cdots & 1 \\ 1 & e^\epsilon & \cdots & 1 \\ \vdots & \vdots & \ddots & \vdots \\ 1 & 1 & \cdots & e^\epsilon \end{bmatrix}.
$$

Its inverse has infinity norm:

$$
\|\mathcal{L}_{\text{LDP2}}^{-1}\|_\infty = \frac{1 + e^{-\epsilon}(4K-3)}{1 - e^{-\epsilon}},
$$

which further inflates the error by nearly a factor of 2 compared to the censored mechanism.

