# OpenReview forum: "Censoring with Plausible Deniability: Asymmetric Local Privacy for Multi-Category CDF Estimation"
_ICLR.cc/2026/Conference — Submitted to ICLR 2026_

### Official Review · Reviewer_4p71 · 2025-10-23

**Soundness:** 1
**Presentation:** 2
**Contribution:** 2
**Rating:** 2
**Confidence:** 3

**Summary:**

This work proposes a mechanism for estimating conditional distribution functions
$P(X \leq x \mid Y=k) \mid k \in \\{1,\dots, K\\}$ from $I$ users' features $(x_i, y_i) \in [0,1] \times \\{1,\dots, K\\}$ in a locally private manner.
Unlike in standard local differential privacy, which requires privacy uniformly for all values, this work seeks a form of asymmetric privacy in which only large values of $x_i$ are considered more sensitive.
An additional challenge this works seeks to address is that it may be hard to define a threshold $t_n$ that divides sensitive and non-sensitive values.

The proposed mechanism largely works as follows:
First, a threshold $t_i$ is randomly sampled from $G = \mathrm{Uniform}(0,1)$ for each user $i$.
Second, their features are mapped into $\mathcal{X} = \\{0,1\\} \times \\{1,\dots, K\\}$ via the corresponding indicator function.
This transformed feature space is thus considered divided into a sensitive region $\mathcal{X}_N = \\{1\\} \times \\{1,\dots, K\\}$, which corresponds to large $x_i$, and a safe region (its complement).
To guarantee privacy for these transformed features, the proposed mechanism (I will forego the indicator function notation here) deterministically maps all elements from
$\mathcal{X}_N$ to $1$ (censoring the categorical feature) and randomly maps elements $(0, k)$ from $X \setminus \mathcal{X}_N$ to either $(0, k)$ or $1$.

This mechanism is shown to be privacy-preserving in the following sense ("ULDP" from prior work):
For sensitive output $1$, it is hard (in the DP sense) to determine the original transformed feature from $\mathcal{X}_N$. No privacy is required for the non-sensitive outputs
in $\\{0\\} \times \\{1,\dots, K\\}$.

Using the post-processing property of DP, the authors then present some MLE-based method for recovering
They further derive error bounds, which characterize the expected error in estimating $P(X \leq x \mid Y=k)$, averaged over all $k$, when randomly sampling $x$ from the threshold-generating distribution $G$.

Finally, the method is numerically evaluated on its ability to recover some simple hand-crafted CDFs.

**Strengths:**

* The mechanism for privately mapping from transformed discrete features to CDFs is well-founded and improves upon prior work
* The theoretical evaluation of error bounds / asymptotic properties is extensive, and makes for a much better contribution than pure empirical evaluation of the privacy--utility trade-off
* The considered problem: Only protecting sensitive values, without knowing exactly which range of values is sensitive, is novel and well-motivated
* The outline in Section 1.2 helps significantly in following the paper
* Limitations are openly and extensively discussed

**Weaknesses:**

## Main Weakness
The main issue I see with the paper is that it **does not address the actual problem of performing ULDP with unknown sensitive ranges** and it is **not clear what form of privacy guarantee is made w.r.t. the original continuous features**.
The underlying cause is that the privacy analysis only starts *after* we have decided on thresholds $t_i$, instead of analyzing the entire mechanism.

Generally speaking, when analyzing the privacy of a composite mechanism $M = M_1 \circ M_2$, it is not sufficient to only analyze the privacy of $M_1$ in the co-domain of $M_2$.

More concretely, this work essentially assumes that whatever thresholds we sample do in fact accurately separate sensitive and non-sensitive value, and provides its guarantees under this assumption.
But there being no fixed "correct" thresholds is precisely what motivated the research in the first place.
To me, it seems like the work is trying to model some prior belief about the correct thresholds for each user (here: an uninformative, uniform prior). But it does not model at all what happens when there is a mismatch between the unknown thresholds $t_i^*$ and the sampled threshold $t_i$. For example, we can probably say that being in the top $0.1\\%$ of earners is sensitive. But with the proposed mechanisms, there is a
slightly smaller than $0.1\\%$ **chance that this sensitive information is leaked without any privacy protection at all**.
It appears like actually solving the considered problem requires defining some $(\epsilon,\delta)$-style relaxed ULDP notion that models the inherent uncertainty of the sensitive range and resulting failures of providing ULDP.

## Other Weaknesses
Presentation:
* The abstract highlights "extensive numerical experiments" as a core contribution. However, these results only appear in the appendix (and are not particularly extensive either). It seems like the authors did not put enough care in actually writing their work for the conference format
* While the overall storyline (see Outline in Section 1.2) is coherent, the individual sections 3-5 are poorly structured. They are essentially just a stream of prose, interspersed with equations. This makes it hard to deduce the main arguments the authors are trying to make. The reading flow could be much improved by including more Proposition/Definition/Remark etc. environments.
* The algorithm for computing CDFs from the mechanism's output is not sufficiently discussed. The authors just point at some existing paper, which puts an unnecessary burden on the reader.

Other:
* The method is only evaluated on a handful of hand-crafted CDFs instead of real-world data. One could compare the private CDF with the empirical CDF to get better insights into the real-world utility of the mechanism.
* While I understand that pure DP work is regularly published at ML conferences, the authors should at least make an effort to explain how their work is relevant for machine learning purposes
* It's somewhat problematic that the main utility metric (expected error in CDF) is directly coupled with the threshold distribution $G$ of the mechanism. Example: Consider threshold distribution $G = \delta_{0.5}$. Then the metric would only measure how well the true CDF is approximated at $x=0.5$, completely ignoring that it is an extremely poor match (a heaviside step function) everywhere else.
* Appendix F only shows that the mechanism improves upon randomized response, but not that it improves upon the "utility-optimized randomized response mechanism" mentioned in Section 3.2.

## Minor comments

* The subscripts $N$, $I$, $S$, $P$ in Definition 3 are somewhat confusing. Wouldn't using $S$ ("sensitive") and $N$ ("non-sensitive") for both the domain and co-domain be sufficient.
* Definition 1 speaks of "randomized algorithms" whereas Definition 2 speaks of "randomized mechanisms".
* In Definition 1, it is not clear that datasets $S$ are composed of elements of $X$, which (for a reader without a DP background) would make it hard to tell the difference to Definition 2
* "Sensitivity" already has a well-defined meaning in the differential privacy context. I would suggest using another word for "being sensitive".
* "while allowing exact outputs" in (ll. 160) is somewhat imprecise. ULDP can also be fulfilled if we do not return the exact value, but some deterministic/invertible transformation of it, right?
* Using $\mathcal{Y}$ for the co-domain of the mechanism and $Y$ for the categorical part of the domain is slightly confusing.
* Introducing $\Delta_i$ from Section 3.3 earlier would have made the one-hot construction in Section 3.2 much easier to follow



## Conclusion
If this work were to only operate in the usual ULDP framework, it would be a relatively straight-forward work that proposes a somewhat improved mechanism with nice theoretical analysis but somewhat lacking presentation and experimental evaluation. This would put it somewhere in borderline territory.

However, the authors open up a much more interesting problem: How to adapt ULDP mechanisms when the true separation into "sensitive" and "safe" values is unknown. A solution to this problem could be much more impactful and be of broader interest for the privacy commmunity.
Unfortunately, their proposed method fails to adequately address this problem and does not provide sound privacy guarantees (see "Main Weaknesses" above).
I would encourage the authors to revise their manuscript, and maybe try to explicitly model the interaction between uncertain "sensitive ranges" and the randomly sampled thresholds, for a sound and much stronger submission.

**Questions:**

/

---

> ### Author Response · Authors · 2025-11-24
>
> We sincerely appreciate the time and effort you have invested in reviewing our paper. Your feedback is invaluable to us, and we address each of your concerns below. Following your advice, we have carefully revised the manuscript, with all modifications highlighted in blue. Should you have any further questions, please do not hesitate to ask.
>
> **For Weakness**
>
> 1. "The main issue.... ULDP"
>
> Response: Thank you for the thoughtful comment. Our work aims to address ULDP when the sensitive region is unknown, through a two-stage mechanism. We clarify the privacy guarantee and the role of threshold randomization below.
>
> We begin with the original data domain $(X_i, Y_i)$, where $X_i \in [0,1]$ is a continuous attribute and $Y_i \in {1,\dots,K}$ is a discrete covariate. Since the sensitive region is user-dependent and unknown, we introduce a random threshold $t_i \sim \mathrm{Uniform}[0,1]$ to define a latent sensitive status $B_i = \mathbf{1}{X_i \le t_i}$. This induces a derived representation $(B_i, Y_i) \in {0,1} \times {1,\dots,K}$, which is then privatized using a ULDP mechanism.
>
> The ULDP guarantee applies to this derived space, not directly to the original $(X_i, Y_i)$. We do not claim that the raw continuous input is ULDP-protected. Instead, we protect a coarsened representation that is sufficient for estimation and inherently less informative. Under a fixed ULDP privacy budget, protecting this transformed representation is arguably more acceptable to users than directly protecting raw data.
>
> Regarding the concern that “sensitive information is leaked without protection”, this does not occur under ULDP. Although sensitive users report truthfully, protection arises from the randomized responses of non-sensitive users. Because some non-sensitive users output sensitive values with positive probability, sensitive users are not identifiable in aggregate.
>
> To illustrate, consider a binary response taking value “A” (sensitive) or “B” (non-sensitive). Without privacy, observing “A” reveals sensitivity with certainty.
>
> Under an $\varepsilon$-LDP mechanism, all users randomize. Let $p = \frac{1}{1+e^\varepsilon}$ be the flip probability. The posterior probability of sensitivity upon observing “A” is
>
> $$
> \Pr(\text{sensitive} \mid \text{response} = A)
> = \frac{\pi_s (1 - p)}{\pi_s (1 - p) + (1 - \pi_s) p}.
> $$
>
> Here $\pi_s$ denotes the prior fraction of sensitive users.
>
> Under ULDP, only non-sensitive users randomize. A non-sensitive user reports “A” with probability $p$, and a sensitive user reports “A” deterministically. The posterior becomes
>
> $$
> \Pr(\text{sensitive} \mid \text{response} = A)
> = \frac{\pi_s}{\pi_s + (1 - \pi_s) p},
> $$
>
> which is strictly less than 1 for any $p > 0$ and $\pi_s < 1$. Thus, even deterministic reporting by sensitive users does not cause full disclosure.
>
> In our setting, the threshold $t_i$ is private and independently sampled. For an extreme value of $X_i$ (for example, the top 0.1% of the domain), the probability of being labeled sensitive is at most 0.001, and this sensitive flag is then ULDP-protected. The overall procedure consists of a randomized sensitive-status determination followed by ULDP protection.
>
> 2. "The abstract .... format."
>
> Response: Thank you for the comment. We have conducted several numerical experiments to verify and further investigate our methodologies, including:
>
> * the divide-and-conquer ICM algorithm (Appendix D.1),
> * comparisons of alternative oracle capping mechanisms (Appendix D.2),
> * evaluations of the ULDP CDF estimator and its prediction error (Appendix D.3),
> * and relative error analysis (Appendix D.4).
>
> Due to space limitations, these results do not all appear in the main text. However, each experiment is now properly referenced in Section 5.
>
> 3. "While .... environments."
>
> Response: Thank you for your suggestions. We have revised the manuscript accordingly to provide a clearer and more consistent presentation.
>
> 4. "The algorithm .... the reader."
>
> Response: Thank you for your suggestion. We have added a brief introduction to the ICM algorithm and included the ULDP CDF estimation algorithm in the revised manuscript to improve the overall presentation.
>
> 5. "The method... mechanism."
>
> Response: We agree that comparison with empirical CDFs on real data is important for assessing practical utility. Appendix E.2 presents a real-data study based on the government salary dataset. In that section, we explicitly compare the private CDFs with the empirical CDFs, which shows that on real-world data the released CDFs track the empirical CDFs closely.

---

> ### Author Response · Authors · 2025-11-24
>
> 6. "While ..... applications."
>
> Response: Our contribution is directly relevant to ML along two complementary axes. First, we propose a deployable client-side ULDP data-collection mechanism (binary thresholding with asymmetrically censored randomized response) that gives per-user local privacy control without predefining sensitive regions and achieves higher utility than standard LDP. This component is independently useful for federated and on-device telemetry as well as surveys. Second, we provide a statistically principled estimator that recovers class-conditional joint CDFs from the privatized reports. These distributional summaries are routinely used in ML for calibration and threshold selection, ROC/PR and cost-sensitive decision rules, monitoring and shift detection, and fairness diagnostics based on quantities such as $P(X > t \mid Y = k)$.
>
> 7. "It .... else."
>
> Response: We clarify that the distribution $G$ is user-designed. Our default choice is the uniform distribution on $[0,1]$, and when the support is $[0,1]$, the $L_{p,G}$ norm is equivalent to the usual $L_p$ norm. If the user specifies a singular $G$, as in your example, then $G$ emphasizes only certain points of the CDF. In that situation, the $L_{p,G}$ norm effectively reduces to an $\ell_2$ norm on a finite set in Euclidean space, measuring discrepancies only at the points of interest specified by the user.
>
> 8. "Appendix F .... Section 3.2."
>
> Response: We agree that a direct comparison to the utility-optimized randomized-response mechanism is interesting. However, such a comparison would require defining a fixed sensitive region on the continuous domain, which is incompatible with our setting. Our method explicitly avoids pre-specifying the sensitive region and instead incorporates per-user random thresholds to capture uncertainty in what is considered sensitive.
>
> 9. "The .... sufficient?"
>
> Response: We agree that the notation in Definition 3 can be difficult to parse in isolation. The subscripts $(N, I, S, P)$ are inherited from the original paper that introduced ULDP, and we kept them to remain consistent with that definition and to make cross-referencing between the two works easier for readers.
>
> 10. "Definition 1 .... mechanisms."
>
> Response: Thank you for pointing this out. We agree that the terminology should be consistent. In the revision, we will unify all references under the term “mechanism” to avoid confusion. Specifically, we will update Definition 1 to use “randomized mechanism,” matching the wording in Definition 2 and the rest of the paper.
>
> 11. "In Definition 1.... Definition 2."
>
> Response: Thank you for pointing this out. We agree that the terminology should be consistent. In the revision, we will unify all references under the term “mechanism” to avoid confusion. Specifically, we will update Definition 1 to use “randomized mechanism,” matching the wording in Definition 2 and the remainder of the paper.
>
> 12. "‘Sensitivity’.... ‘being sensitive’."
>
> Response: We acknowledge the overlap in terminology. The term “sensitive” has a specific meaning in differential privacy (as in “global sensitivity”), which differs from its usage in the ULDP framework, where it refers to whether a value belongs to a protected category. In our paper, we follow the ULDP usage to remain consistent with the original definition. Since we do not use “sensitivity” in the DP sense (such as $\ell_1$ sensitivity of functions) anywhere in our work, we believe the likelihood of confusion is limited.
>
> 13. "The phrase .... correct?"
>
> Response: Yes, ULDP can still be satisfied even if the mechanism outputs a deterministic and invertible transformation of the true value rather than the value itself. However, when ULDP conditions already permit releasing the raw output exactly, there is no practical benefit to adding such a transformation. Our phrasing “while allowing exact outputs” was intended to emphasize that, unlike standard LDP, ULDP does not require randomization on non-sensitive values, making exact release permissible within the definition. We will revise the text to clarify this interpretation.
>
> 14. "Using.... confusing."
>
> Response: We follow the original ULDP convention: calligraphic letters denote spaces (for example, $\mathcal{X}$, $\mathcal{Y}$), while roman capitals denote random variables (for example, $X$, $Y$). In our setting, $Y \in {1, \ldots, K}$ is the categorical variable; the mechanism’s output range is a distinct space $\mathcal{E}$ (one-hot outputs), which prevents overloading $Y$.
>
> 15. "Introducing $\Delta_i$... to follow."
>
> Response: Thank you for the advice. $\Delta_i$ is introduced at the end of Section 3.2 and is not defined earlier. We have added a diagram (Figure 1) to clarify its role.

---

> ### Comment · Reviewer_4p71 · 2025-11-27
>
> Thank you for the extensive rebuttal. Points 2-15 adequately address all the "other weaknesses" and "minor comments" in my review above.
>
> Point 1 partially addresses my main criticism insofar as it demonstrates that there is no complete privacy leakage, even when there is a mismatch between the true, unknown sensitive range, and the sampled sensitive range.
> **However, the core issue remains: A detailed privacy analysis is only provided w.r.t. the sampled sensitive ranges / "latent sensitive status".**
>
> I have increased my score to "borderline reject" for now.
>
> I would be willing to further increase my score to "borderline accept", if the authors were to explicitly state that the mechanism is ULDP w.r.t. the latent/sampled sensitive status. Such an explanation could, for example, be added to the end of l.256 or within an additional Remark environment at the end of page 5.
> This change would help to prevent follow-up works from misinterpreting the results in this paper.

---

> > ### Author Response · Authors · 2025-11-28
> >
> > Thank you for agreeing to raise the score to “borderline accept.” We appreciate the suggestion to add a remark clarifying that the ULDP condition is applied on the latent sensitive-status domain, and we have incorporated this clarification into the manuscript (see Remark 2 at the end of page 5). Your constructive feedback has substantially improved the clarity of our work, and we are glad that this discussion has resolved the remaining concerns.

---

### Official Review · Reviewer_iszU · 2025-10-28

**Soundness:** 3
**Presentation:** 3
**Contribution:** 3
**Rating:** 6
**Confidence:** 4

**Summary:**

This paper proposes a new mechanism for Utility-Optimized Local Differential Privacy (ULDP) that enables asymmetric privacy protection—censoring potentially sensitive responses while maintaining plausible deniability. Instead of predefining a sensitive region, the mechanism relies on a direction of sensitivity (e.g., higher income or debt values are more sensitive). It combines deterministic thresholding with randomized response to protect sensitive values while retaining useful information about nonsensitive ones. The authors define the Asymmetrically Censored Randomized Response (ACRR) mechanism. Then they develop a maximum likelihood estimator (MLE) for CDF estimation under ULDP and prove $L^2$ and uniform consistency (rates $n^{-1/3}$).

**Strengths:**

- Novel conceptual contribution: The idea of asymmetric censoring with plausible deniability is both original and practically motivated. It moves beyond uniform privacy budgets by allowing directional sensitivity—aligning privacy protection more closely with real-world data semantics (e.g., “high debt” being sensitive).
- Solid theoretical foundation: The asymptotic analysis (Theorems 1–3) is rigorous and builds on advanced shape-constrained inference and current-status data literature. The proof techniques are carefully adapted to handle ULDP’s non-differentiabilities.
- Strong connection between privacy and statistics: The paper effectively bridges differential privacy and nonparametric inference, showing that local privacy mechanisms can be interpreted through censoring frameworks common in survival analysis.
- Practical mechanism design: The proposed ACRR is interpretable, easy to implement, and avoids arbitrary sensitive-region definitions—a major usability improvement over earlier ULDP schemes.

**Weaknesses:**

- Please use \cite, \citep, and \citet correctly. Note that the form of \cite can vary between different Latex template.
- Line 114: The symbol $K$ appears for the first time here but is not defined. Please define $K$ at its first occurrence.
- About the definition of ULDP:
	- The initial presentation of the ULDP definition is somewhat difficult to follow and would benefit from a more intuitive explanation or an illustrative example. Since ULDP is not yet a widely adopted concept, I strongly suggest adding a short, concrete example (e.g., a toy scenario) to help readers quickly grasp what the mechanism is doing.
	- The definition itself is quite interesting. Taking the survey protocol in Appendix A as an example, the plausible deniability comes from some low-debt participants occasionally reporting “Yes,” thereby concealing the true high-debt participants. However, from the perspective of high-debt individuals, the only valid output is “Yes.” This asymmetry could lead to survey refusal or misreporting if respondents are not well informed about the mechanism. My question is: if such refusal occurs, can the current framework naturally accommodate it—for example, by allowing some users (possibly those with high debt, i.e., data-dependent) or a predefined subset of users to instead follow a standard LDP mechanism? This issue has been discussed in the literature in the context of heterogeneous privacy requirements, where different data records may demand different protection levels. See, for example [1,2].

[1] Versatile Differentially Private Learning for General Loss Functions.

[2] Locally Private Estimation with Public features.

**Questions:**

See weaknesses. I would be happy to raise my score if the questions were discussed in depth.

---

> ### Author Response · Authors · 2025-11-24
>
> We sincerely appreciate the time and effort you've invested in reviewing our paper. Your feedback is invaluable to us, and we have addressed each of your concerns below. Following your advice, we have carefully revised the manuscript, with all modifications highlighted in blue. Should you have any further questions, please do not hesitate to ask.
>
> **For Weakness**
> 1. "Please use .... template."
>
>
> - Response: Thank you for your correction. We have revised the citation format accordingly.
>
>
> 2. "Line 114: The .... occurrence. "
>
>
> - Response: Thank you for your suggestion. We have added it.
>
>
> 3. "The initial ... doing."
>
>
> - Response:
> We agree that the initial presentation of the ULDP definition may be difficult to follow, especially for readers unfamiliar with this relatively recent concept. To address this, we included Appendix~A, which provides a concrete toy example illustrating how ULDP works in a simple setting. This example walks through a binary response scenario and highlights the asymmetry between the treatment of sensitive and non-sensitive values—central to understanding ULDP.
>
> We believe Appendix~A serves precisely the purpose of offering intuition behind the definition. In the revised version, we will make this clearer by explicitly referencing the appendix when ULDP is first introduced in the main text, and we will consider adding a brief summary or visual in the main body to help orient readers earlier.
>
> 4. "The definition itself ......"
>
>
> - Response: We agree that the asymmetry in ULDP—where sensitive users report truthfully while only non-sensitive users randomize—can appear unintuitive and may raise concerns about respondent trust. However, this asymmetry  offers strong plausible deniability, as protection arises from the randomized responses of non-sensitive users.
>
> To make this concrete, consider a binary-response survey where “A” is a sensitive response and “B” is non-sensitive. Under no privacy mechanism, a user reporting “A” is deterministically identified as sensitive. Under standard $\varepsilon$-LDP, all users randomize: each reports the opposite value with probability $p = 1/(1 + e^\varepsilon)$. Then, by Bayes' rule, the posterior probability of a respondent being sensitive given an “A” response is
> $
> \Pr[\text{sensitive} \mid \text{response} = A] = \frac{\pi_s (1-p)}{\pi_s (1-p) + (1 - \pi_s)p},
> $
> where $\pi_s$ is the prior probability of being sensitive.
>
> In ULDP, only non-sensitive users randomize. A sensitive user reports “A” truthfully, while a non-sensitive user reports “A” with probability $p$. Then the posterior is
> $
> \Pr[\text{sensitive} \mid \text{response} = A] = \frac{\pi_s}{\pi_s + (1 - \pi_s)p},
> $
> which is again strictly less than 1. Thus, even though sensitive users report deterministically, their responses remain protected by the randomized behavior of the non-sensitive population.
>
> As for the possibility of refusal or distrust due to this asymmetry, we agree that this is a valid concern, especially in settings where users do not fully understand the mechanism. The current framework assumes a homogeneous user population, where all participants accept and follow the same ULDP protocol. We do not model user-specific refusal or allow per-user deviations from the mechanism.
>
> We are aware of the literature on heterogeneous privacy settings, where different users or data records may require different protection levels. Allowing, for example, some users to opt into standard LDP while others follow ULDP would require extending the framework to support such heterogeneity—possibly with additional inference safeguards and utility adjustments. Nevertheless, this is an interesting extension, and we have added these works to the discussion in the introduction.

---

> > ### Comment · Reviewer_iszU · 2025-11-26
> >
> > The authors have clearly addressed my question regarding the clarity of the ULDP definition. I believe the paper reaches the acceptance threshold, and I have accordingly raised my score.

---

> > > ### Author Response · Authors · 2025-11-26
> > >
> > > Thank you very much for your acknowledgment of our response. We are glad that we have addressed your concern, and we appreciate your decision to raise your score. We will ensure that the final version of the paper reflects the clarifications and improvements discussed. If you have any further questions or suggestions, please do not hesitate to let us know.

---

### Official Review · Reviewer_J9hd · 2025-10-31

**Soundness:** 2
**Presentation:** 3
**Contribution:** 2
**Rating:** 4
**Confidence:** 4

**Summary:**

The paper proposes a new mechanism in the Utility-Optimized Local Differential Privacy framework that enables censoring with plausible deniability for sensitive data collection. It targets settings with asymmetric sensitivity (e.g., large numeric responses are more private than small ones, and categorical context can be identifying) by selectively withholding identifying details when a response likely signals sensitive content. Unlike prior approaches, the mechanism does not require pre-specifying which values are sensitive, improving practicality and adaptability; it also applies under standard symmetric LDP, where censoring can still lower privacy cost. The authors provide theoretical guarantees, including uniform consistency and pointwise weak convergence, and corroborate the approach with extensive numerical experiments demonstrating favorable utility–privacy trade-offs.

**Strengths:**

The paper is technically sound and the proofs appear correct. The paper provides both theoretical guarantees and empirical results.

**Weaknesses:**

**Major concerns**

- Missing baselines: The paper lacks comparisons against baseline methods.

- Utility bounds and privacy dependence: The stated utility guarantees (Theorems 1 and 2) do not appear to depend on the privacy parameter $ \varepsilon $. It would strengthen the work to provide utility bounds for the private algorithm that make the $ \varepsilon $-dependence explicit.

- Absent privacy proofs: I could not find a formal proof of the mechanisms’ privacy guarantees (eg. ACRR in defn. 4). Please include precise statements (with assumptions and neighboring relations) and complete proofs or clear references.

- Meaning of “adaptive” censoring: The claim “a flexible ULDP mechanism that *adaptively* censors potentially sensitive responses without requiring a predefined sensitive region” needs clarification. What quantities are adapted, based on which statistics or at what granularity?

**Minor issues**

- The paper never defines “censoring.” A formal definition would be helpful.

**Questions:**

Please see the weakness section.

---

> ### Author Response · Authors · 2025-11-24
>
> We sincerely appreciate the time and effort you've invested in reviewing our paper. Your feedback is invaluable to us, and we have addressed each of your concerns below. Following your advice, we have carefully revised the manuscript, with all modifications highlighted in blue. Should you have any further questions, please do not hesitate to ask.
>
> **For Weakness**
> 1. "Missing baselines:....."
>
>
> - Response: We agree that including direct baseline comparisons would strengthen the work. To the best of our knowledge, however, there is currently no method that addresses the same problem without requiring a pre-specified sensitive region, so a fully comparable baseline is not available. In Appendix~F, we therefore consider a natural variant of our approach obtained by replacing the ACRR mechanism with a standard LDP mechanism, and compare the resulting perturbation matrix $\mathcal{L}$ in terms of its eigenvalues and $\|\mathcal{L}^{-1}\|_\infty$.
>
> 2. "Utility bounds and privacy dependence:...."
>
>
> - Response:  We agree that making the dependence on $\varepsilon$ explicit is helpful. In our analysis, the linear operator $\mathcal{L}$ is defined in terms of $\varepsilon$ and $K$ (line 335). Its smallest eigenvalue is $1 - e^{-\varepsilon}$, and
> $
> |\mathcal{L}^{-1}|_\infty ;=; \frac{1 + e^{-\varepsilon}}{1 - e^{-\varepsilon}},
> $
> which does not depend on $K$ (an major advantage of our method). The constants in the utility bounds of Theorems 1 and 2 therefore depend on $\varepsilon$ only through this factor. We will revise the theorem statements to make this $\varepsilon$-dependence explicit.
>
> 3. "Absent privacy proofs:...."
>
>
> - Response:  In line 254 of original manuscript, we mentions that the map in ACRR in special case of the utility-optimized randomized response mechanism from Murakami \& Kawamoto, this approach provides $(\mathcal{X}_N,\{e_{K+1}\},\epsilon)$-ULDP.
>
> 4. "Meaning of ``adaptive'' censoring:....."
>
>
> - Response: We thank the reviewer for the request for clarification. By adaptive censoring we mean that the mechanism determines which responses to censor directly from the observed statistics of the data, without requiring any predefined bins of the domain or any preset sensitive region. The censoring rule is therefore derived automatically from the empirical distribution rather than fixed in advance. We have revised the text to make this meaning explicit while keeping the term adaptive.
>
> 5. "The paper never ..."
>
>
> - Response: In general, “censoring’’ refers to situations where some variables are only partially observed or not observed at all. In our setting, censoring specifically means that respondents who give a sensitive response are excused from providing potentially identifying demographic information, so their covariates are unobserved.
>
>
> **For Questions**
>
> See W1-W5.

---

### Official Review · Reviewer_HyAJ · 2025-10-31

**Soundness:** 3
**Presentation:** 2
**Contribution:** 2
**Rating:** 4
**Confidence:** 3

**Summary:**

This paper proposes a novel asymmetric local differential privacy mechanism under the Utility-Optimized Local Differential Privacy (ULDP) framework.

The key contribution is a censoring mechanism with plausible deniability, designed to protect users whose responses may implicitly reveal sensitive information (e.g., large numerical values). Unlike previous approaches, the method does not require predefined sensitive regions, instead using a direction of sensitivity (e.g., “larger means more sensitive”).

The work represents the first asymptotic analysis of CDF estimation under ULDP with multi-category prediction, bridging privacy-preserving data collection and nonparametric estimation theory.

**Strengths:**

1.  **Originality**: Introduces asymmetric local privacy with plausible deniability—protecting sensitive values adaptively without predefining a sensitive region; Bridges ULDP and classical CDF estimation theory, leveraging Chernoff-type asymptotics.
2.  **Quality**: Rigorous proofs of consistency and weak convergence and nontrivial extension of LDP results (Liu et al., 2024) from single to multi-category settings.

**Weaknesses:**

1.  The estimator’s runtime grows superlinearly with sample size and linearly with the number of categories K.
2.  The “capping at 1” rule for CDF may distort tail behavior; a quantitative analysis of this effect would strengthen the paper.
3.  The $O_p(n^{-1/3})$ rate is slower than the parametric $O_p(n^{-1/2})$ rate; discussion of possible improvements (e.g., semi-parametric approaches) would be valuable.

**Questions:**

1.  Can the mechanism handle non-monotonic sensitivity, where both small and large values are sensitive?
2.  Could jointly estimating sub-CDFs under a sum-to-one constraint mitigate boundary bias?
3.  How would the estimator generalize to multivariate continuous data $X \in \mathbb{R}^d$ ?
4.  Can the mechanism be adapted to streaming or online ULDP settings?

---

> ### Author Response · Authors · 2025-11-24
>
> We sincerely appreciate the time and effort you've invested in reviewing our paper. Your feedback is invaluable to us, and we have addressed each of your concerns below. Following your advice, we have carefully revised the manuscript, with all modifications highlighted in blue. Should you have any further questions, please do not hesitate to ask.
>
> **For Weakness**
> 1. "The estimator's .... $K$."
>
> - Response:
>  Thank you for raising the issue of computational time, which is a concern of us and was discussed in Appendix E.1. Truly linear-time behavior is unrealistic in our setting: even computing the empirical CDF from raw data requires a sorting step, which is already ($O(n \log n)$). Our optimization routine uses the Iterative Convex Minorant algorithm (Groeneboom \& Jongbloed, Nonparametric Estimation under Shape Constraints, 2014, Sec. 7.3), where each iteration is linear in the sample size but the number of iterations grows with (n), leading to super-linear overall runtime.  This method is already magnitudes faster than EM algorithm To mitigate this, we adopt a divide-and-conquer strategy and empirically observe acceptable running times for sample sizes up to ($10^8$).
>
> 2. " The ``capping at 1''... paper."
>
> - Response: Allowing the estimated CDF to exceed 1 would align more directly with the error bounds in Theorems 1–3, but then the result is no longer a proper CDF and is harder to interpret or use in practice. To quantify the impact of capping, Appendix E.2 compares our capped estimator with an oracle version that uses the true CDF value at 1; the observed differences in tail behavior are small in all settings we considered.
>
> 3."The $O_p(n^{-1/3})$ ... valuable."
>
> - Response: Thank you for your suggestions. As you pointed out, although our result provides the first theoretical guarantee, its convergence rate is slower than the non-DP optimal parametric rate $\mathcal{O}_p(n^{-1/2})$. For cube-root estimators, there exist potential improvements such as the smoothed estimator [1] and the federated estimator [2], which refine the cube-root rate to $\mathcal{O}_p(n^{-1/3})$ and $\mathcal{O}_p(S^{-1/2}n^{-1/3})$ respectively, where $S = o_p(n^{-1/6}/\log^{6/5} n)$. We observe similar improvement via the later approach in
> Section E.1. While incorporating such refinements into our ULDP framework is interesting, it poses substantial challenges, and we leave this direction for future work. Following your suggestion, we have summarized these potential improvements in Section~5.
>
> **For Questions**
>
> 1. "Can the mechanism ....sensitive?"
>
> - Response: Our mechanism, as formulated, assumes monotone sensitivity and does not naively handle patterns where both small and large values are sensitive but mid-range values are not. One possible workaround is to apply an invertible transform that folds the sensitive regions into a single interval where the sensitivity is monotone. For example, one can map $[0,\delta]$ to $[1-2\delta,1-\delta]$ and $[\delta,1-\delta]$ to $[0,1-2\delta]$ by a suitable shift, so that small and large original values are treated similarly after transformation. Our mechanism is then applied to $g(X)$, and we report $g^{-1}$ of the released outputs. This involves additional modeling choices however.
>
> 2. "Could jointly ... bias?"
>
> - Response: Thank you for your question. Yes, our algorithm outputs a joint estimator for all sub-CDFs under the sum-to-one constraint. One of our main contributions is designing a ULDP algorithm that preserves the relationships among the sub-CDFs, rather than estimating them separately. The main theoretical results are also established for these joint estimators.
>
> 3. "How would ... $X \in \mathbb{R}^d$"
>
> - Response: Thank you for your questions. The answer is yes, but the approach becomes substantially more complicated. For estimating a multivariate distribution function, for example $X = (X_1, X_2) \in \mathbb{R}^2$, one may still consider observations of the form $(\Delta_1, \Delta_2) = (1_{x_1 \leq T_1}, 1_{x_2 \leq T_2})$ with appropriately designed thresholds $T_1, T_2$. The likelihood function can be derived in a similar manner. However, the computation becomes challenging, as the ICM algorithm is not straightforward to implement in a multivariate setting. For these reasons, we consider this direction beyond the scope of our current work.

---

> ### Author Response · Authors · 2025-11-24
>
> 4. "Can ... settings?"
>
> - Response: Thank you for your question. In the online setting, our current implementation for solving the likelihood function is not an online algorithm, so extending it directly to an online ULDP framework is nontrivial unless alternative numerical approaches are introduced. For streaming data, one may apply our algorithm to each incoming data batch and then average the resulting estimators to obtain a final ULDP distribution estimator. The theoretical properties of this approach remain unknown, but it resembles the divide-and-conquer framework discussed in Section~E of the original version, and we expect the numerical performance to be comparable.
>
> - Reference
>
>
> [1] Groeneboom, P., Jongbloed, G., \& Witte, B. I. (2010). Maximum smoothed likelihood estimation and smoothed maximum likelihood estimation in the current status model. Annals of Statistics, 38(1), 352-387.
>
>
> [2] Shi, C., Lu, W., \& Song, R. (2018). A massive data framework for M-estimators with cubic-rate. Journal of the American Statistical Association, 113(524), 1698-1709.

---

> ### Comment · Reviewer_HyAJ · 2025-11-26
>
> Thanks for the clarification. I will increase my score.

---

> > ### Author Response · Authors · 2025-11-26
> >
> > Thank you very much for your acknowledgment of our response. We are glad that we have addressed your concern, and we appreciate your decision to raise your score. We will ensure that the final version of the paper reflects the clarifications and improvements discussed. If you have any further questions or suggestions, please do not hesitate to let us know.

---

### Author Response · Authors · 2025-12-03
**Author Final Remarks**

We thank the reviewers for their careful evaluation of our submission and for the constructive discussion. Since the reviewers are no longer allowed to participate further in the forum, we would like to briefly summarize how the discussion evolved, how we revised the manuscript, and how the reviewers' views changed over time.

Of the three reviewers who participated in the discussion before the unexpected interruption, all improved their assessment significantly (by at least two points) after our clarifications and revisions. In their later comments, these reviewers explicitly stated that their main concerns had been resolved or substantially alleviated, and their written assessments became much more positive than what is reflected in the original pre-discussion scores.

There was broad agreement in the discussion that the core idea of the paper is novel and practically relevant. The proposed asymmetric ULDP mechanism with plausible deniability for multi-category CDF estimation was recognized as a nontrivial contribution, well motivated by realistic data-collection scenarios in which only some categories are sensitive. Reviewers also appreciated that we provide both formal guarantees and empirical evaluation, and the conversation naturally shifted from questioning the contribution itself to clarifying its precise assumptions and guarantees.

The main technical concerns raised in the initial reviews were directly addressed in our rebuttal and follow-up comments. Because the sensitive region is not explicitly specified, we clarified the two-stage construction (coarse discretization followed by the ULDP mechanism) and made explicit how this avoids deterministic disclosure of sensitive information while preserving strong plausible deniability. Regarding the asymmetry of the mechanism, we explained why sensitive and non-sensitive categories are treated differently, how this compares to standard symmetric ULDP mechanisms, and why this design is appropriate for the applications we target. We also elaborated on the privacy–utility trade-offs, clarifying the lack of directly comparable prior methods with the same ULDP and CDF focus, with pointers to the extended discussion and experiments in the appendix.

We also made substantial improvements to readability and presentation based on the reviewers’ suggestions. We refined the definitions of sensitive and non-sensitive categories, the ULDP mechanism, and the censoring operator, and we cleaned up inconsistent notation. We reorganized the early sections so that intuition and examples appear before the technical material, and we improved cross-references between the main text and the appendix. In their later comments, reviewers noted that these changes made the mechanism and definitions considerably clearer and that the paper became substantially easier to follow.

We believe the discussion led to a clearer and stronger presentation of a contribution that reviewers acknowledged as novel and meaningful for practical applications. The record of the discussion shows that the reviewers' opinions improved materially beyond what is conveyed by the initial numerical scores, and that their principal technical and presentation-related concerns have been addressed. We respectfully invite the area chair to take these post-discussion developments into account, together with the original reviews, when forming the final recommendation.

---

### Meta-Review · Area_Chair_b9Pu · 2026-01-12

**Summary:**

Many small concerns and one very important concern from 4p71 on the actual privacy guarantee of the paper

**Reviewer Concerns:**

The concern of 4p71 has not been covered. In their final reply, the authors seem to consider that the concerns have been taken in consideration, which is absolutely not the case for the key concern of 4p71

**Reviewer Scores:**

The issue is not the eventual change, but rather that the authors fail to solve the key concern of 4p71

---

### Decision · Program_Chairs · 2026-01-26

Reject